# A small signaling domain controls PPIP5K phosphatase activity in phosphate homeostasis

Pierre Raia [1], Kitaik Lee [1,2], Simon M. Bartsch [3,4], Felix Rico-Resendiz[1], Daniela Portugal-Calisto[5], Oscar Vadas [6], Vikram Govind Panse [5,7], Dorothea Fiedler [3,4] & Michael Hothorn [1] ✉

Inositol pyrophosphates (PP-InsPs) are eukaryotic nutrient messengers. The N-terminal kinase domain of diphosphoinositol pentakisphosphate kinase (PPIP5K) generates the messenger 1,5-InsP$_8$, the C-terminal phosphatase domain catalyzes PP-InsP breakdown. The balance between kinase and phosphatase activities regulates 1,5-InsP$_8$ levels. Here, we present crystal structures of the apo and substrate-bound PPIP5K phosphatase domain from S. *cerevisiae* (ScVip1$^{PD}$). ScVip1$^{PD}$ is a phytase-like inositol 1-pyrophosphate histidine phosphatase with two conserved catalytic motifs. The enzyme has a strong preference for 1,5-InsP$_8$ and is inhibited by inorganic phosphate. It contains an α-helical insertion domain stabilized by a structural Zn$^{2+}$ binding site, and a unique GAF domain that channels the substrate to the active site. Mutations that alter the active site, restrict the movement of the GAF domain, or change the substrate channel's charge inhibit the enzyme activity in vitro, and Arabidopsis VIH2 *in planta*. Our work reveals the structure, enzymatic mechanism and regulation of eukaryotic PPIP5K phosphatases.

Inositol pyrophosphates (PP-InsPs) are eukaryotic nutrient messengers[1]. These intracellular signaling molecules consist of a densely phosphorylated six-carbon *myo*-inositol ring with pyrophosphate groups attached, for example, at position 1 (1-InsP$_7$), position 5 (5-InsP$_7$), or both positions (1,5-InsP$_8$). PP-InsPs regulate cell signaling by a variety of mechanisms, including mediating protein-protein interactions, allosterically regulating enzyme or transporter activities, or post-translationally modifying proteins[2,3].

A central role of PP-InsPs in cellular inorganic phosphate (Pi) homeostasis is conserved among fungi[4–6], protozoa[7], algae[8], plants[9–15], and animals[16–19]. Pi homeostasis is primarily controlled by 1,5-InsP$_8$ binding to PP-InsP sensing SPX receptor proteins[4,5,11,12,14,15,17]. 1,5-InsP$_8$ levels are low under Pi starvation conditions, whereas

1,5-InsP$_8$ accumulates under Pi-sufficient growth conditions or upon Pi resupply[5,13]. As cellular Pi homeostasis and other essential cellular processes appear to be mechanistically linked to 1,5-InsP$_8$ levels, understanding the regulation of 1,5-InsP$_8$ biosynthesis and catabolism is of fundamental importance.

PP-InsPs are generated from InsP$_6$ through a series of phosphorylation steps catalyzed by different small molecule kinases. In yeast, the inositol hexakisphosphate kinase Kcs1[20,21] and in plants, the inositol 1,3,4-trisphosphate 5/6-kinases ITPK1/2[22], preferentially transfer a second phosphate to position 5 of InsP$_6$ or 1-InsP$_7$ to generate 5-InsP$_7$ and 1,5-InsP$_8$, respectively. The diphosphoinositol pentakisphosphate kinases (PPIP5Ks), known as Vip1 in S. *cerevisiae*, Asp1 in *Schizosaccharomyces pombe*[23–25], PPIP5K in amoeba and in animals[16,26–30] or

[1]Structural Plant Biology Laboratory, Department of Plant Sciences, University of Geneva, Geneva, Switzerland. [2]Center for Structural Biology, Center for Cancer Research, National Cancer Institute (NCI), Frederick, MD, USA. [3]Department of Chemical Biology, Leibniz-Forschungsinstitut für Molekulare Pharmakologie, Berlin, Germany. [4]Institute of Chemistry, Humboldt-Universität zu Berlin, Berlin, Germany. [5]Institute of Medical Microbiology, University of Zürich, Zürich, Switzerland. [6]Protein Biochemistry Platform, Faculty of Medicine, University of Geneva, Geneva, Switzerland. [7]Faculty of Science, University of Zürich, Zürich, Switzerland. ✉e-mail: michael.hothorn@unige.ch

VIP/VIH in plants[8,10,31–34], preferentially transfer a phosphate to position 1, to generate 1-InsP$_7$ and 1,5-InsP$_8$, respectively. Consistent with the function of 1,5-InsP$_8$ as messenger in Pi homeostasis in Arabidopsis, *vih1/vih2* loss-of-function mutants have severely reduced 1,5-InsP$_8$ levels and display constitutive Pi starvation responses[10–12].

In addition to the conserved Pi homeostasis-related phenotypes, loss-of-function of PPIP5K enzymes results in altered growth, cell and vacuolar morphology, chromosome segregation, host colonization and DNA damage-induced autophagy[24,25,35,36]. PPIP5K mutants are associated with hearing loss in humans and in mice[37]. In human cell lines, depletion of PPIP5K activity induced broad changes in cell metabolism[38]. In plants, PPIP5K activity has been linked to jasmonic acid and salicylic acid hormone signaling, and to cell wall architecture[32–34,39].

There are several PP-InsP catabolic enzymes, including a family of histidine acid phosphatases located in C-terminal half of PPIP5Ks[24]. Thus, PPIP5Ks are dual-function enzymes that contain two distinct domains with opposing enzymatic activities: An N-terminal kinase domain that preferentially generates 1,5-InsP$_8$ from 5-InsP$_7$[25,26,40], and a C-terminal phosphatase domain with inositol 1-pyrophosphate phosphatase activity[25,29,40–42]. The relative PP-InsP kinase and phosphatase activities in PPIP5K are thought to modulate the cellular levels of 1,5-InsP$_8$[2,10,16].

PPIP5K loss-of-function mutants show altered PP-InsP pools in various eukaryotes[5,8,10,13,24,25,32,34–36,38,43]. PPIP5K knockout mutants have reduced 1-InsP$_7$ and 1,5-InsP$_8$ levels, while the 5-InsP$_7$ substrate of the PPIP5K kinase domain accumulates[5,13,34,44]. 1,5-InsP$_8$ pools can be restored to wild-type like levels in genetic rescue experiments with phosphatase-dead but not with kinase-dead versions of the corresponding PPIP5K enzyme[10,25,35,36,38].

How regulation of PPIP5Ks controls 1,5-InsP$_8$ pools in response to changes in Pi nutrient availability is not well understood. Currently, it is known that the ATP/ADP ratio decreases in response to Pi starvation, which in turn reduces 1,5-InsP$_8$ synthesis by the kinase domains of human PPIP5K1/2 or plant VIH1/2[10,16]. Pi itself has been reported as an inhibitor of the phosphatase domain of human and fungal PPIP5Ks[10,16,40].

The structure and enzymatic mechanism of the human and fungal PPIP5K kinase domains have been well established[27,45,46]. However, currently there is no structural information on the PPIP5K phosphatase domain and its enzymatic mechanism and regulation remain to be elucidated. Based on sequence comparisons, the PPIP5K phosphatase domain has been assigned to the histidine acid phosphatase superfamily[24], specifically to the branch 2, which includes the well-characterized *Escherichia coli* and fungal phytases, as well as glucose-1-phosphate and acid phosphatases[47]. The PPIP5K phosphatase domain contains the catalytic RHxxR motif, which is conserved among histidine acid phosphatases[24]. Histidine acid phosphatases catalyze the hydrolysis of phosphomonoesters via a two-step mechanism. First, the conserved histidine acts as a nucleophile, forming a covalent phospho-histidine intermediate. In a second step, a water molecule hydrolyzes the intermediate, releasing free inorganic phosphate[48]. However, the lack of sequence conservation in the substrate-binding domains within the superfamily precludes predictions about substrate recognition and specificity. A "cryptic" pleckstrin homology (PH) domain located near the conserved RHxxR motif in PPIP5Ks has been hypothesized to play a critical role in substrate binding[26]. PH domains, typically found in signaling proteins, bind phospholipids and molecules derived from phospholipid headgroups, including inositol phosphates and PP-InsPs[49].

Here, we report the structure of a PPIP5K phosphatase domain and characterize its substrate binding and enzymatic mechanisms, its regulation by nutrients and its structural organization within the full-length enzyme.

## Results

### Overall structure of the ScVip1 kinase and phosphatase domains

We first attempted to characterize the structure and catalytic mechanism of the histidine acid phosphatase domain (PD) from plant PPIP5Ks[10,31,32]. However, we were unable to express sufficient amounts of the isolated phosphatase domains from either *Arabidopsis thaliana* VIH1/VIH2 or from *Marchantia polymorpha* VIP1[34] (Supplementary Fig. 1). Therefore, we expressed and purified the phosphatase domains from *S. cerevisiae* Vip1 (ScVip1$^{PD}$, residues 536–1107) as well as its N-terminal kinase domain (ScVip1$^{KD}$, residues 186–522) in baculovirus-infected insect cells (Fig. 1a; see "Methods").

The structure of the ScVip1$^{KD}$ was solved by molecular replacement to a resolution of ~1.2 Å (Supplementary Table 1). Apo crystals of the ScVip1$^{PD}$ initially diffracted to ~3.1 Å. The structure was solved by molecular replacement-single wavelength anomalous diffraction (MR-SAD) on a platinum derivative (Supplementary Table 1). In this structure, a large disordered loop (residues 851–918) was located and subsequently replaced by a short Gly-Ser-Ser-Gly linker. Crystallization of ScVip1$^{PD\ \Delta848-918}$ yielded a second crystal form diffracting at 3.4 Å resolution (Supplementary Table 1).

The structure of ScVip1$^{KD}$ in complex with adenosine diphosphate revealed the typical ATP-grasp fold and substrate binding sites previously seen in other PPIP5K kinase structures[27,46] (Fig. 1a, b). ScVip1$^{KD}$ shares significant sequence and structural homology with HsPPIP5K2 and with the kinase domain of *S. pombe* Asp1 (Supplementary Figs. 2, 3). The C-terminal α11 helix of the ScVip1 kinase domain (residues 498–520) is well defined in our structure (Fig. 1b; Supplementary Fig. 2).

The two apo structures of ScVip1$^{PD}$ revealed a conserved histidine acid phosphatase core with two insertion domains (Fig. 1c). The N-terminus of ScVip1$^{PD}$ (residue 536) and the most C-terminal α9 helix (residues 1079–1094) are well defined by electron density (Fig. 1c). Based on these domain boundaries, we hypothesize that the ScVip1 kinase and phosphatase domains are surrounded by large, unstructured N- and C-terminal extensions, and are connected by a flexible ~15 amino acid linker (Fig. 1a; Supplementary Fig. 2, see below). Plant and human PPIP5Ks contain sequence-diverse linkers of similar length between their kinase and phosphatase domains (Supplementary Fig. 2). An unexpected metal binding site occupied by a Zn$^{2+}$ ion is formed by His651 and His1064 from the histidine acid phosphatase core, and by Cys793 and His836 from the α-helical insertion domain (Fig. 1c; Supplementary Fig. 2; see below).

### A PPIP5K unique GAF domain

Next, we performed structural homology searches with ScVip1$^{PD}$ as a search model using the DALI web server (http://ekhidna2.biocenter.helsinki.fi/dali/)[50]. We found that ScVip1$^{PD}$ shares the core phosphatase domain fold and the α-helical insertion domain with bacterial, plant and human histidine acid phosphatases (Fig. 2a; Supplementary Fig. 4). The relative orientation of the α-helical insertion domain is conserved among histidine acid phosphatases with very different substrate preferences (Fig. 2a), including phytases that hydrolyze InsP$_6$ (Fig. 2b). We located a second insertion domain specific to PPIP5Ks in a position normally occupied by the much smaller β1-insert in other histidine acid phosphatases (Fig. 2a). Analysis with DALI revealed a weak structural homology of this second insertion domain with bacterial and human GAF[51] or PAS[52] domains involved in light, metabolite and cyclic nucleotide sensing[53,54] (Fig. 2c; Supplementary Fig. 4). The GAF domain is conserved among fungal, plant and animal PPIP5Ks (Supplementary Fig. 2). The central β-sheet of the ScVip1$^{PD}$ GAF domain is reminiscent of the inositol phosphate/inositol polyphosphate binding surface previously identified in PH domains, but its topology is different (Fig. 2c)[55,56].

### ScVip1$^{PD}$ specifically hydrolyses 1,5-InsP$_8$

1,5-InsP$_8$ is the preferred substrate for the isolated ScVip1 phosphatase domain in quantitative nuclear magnetic resonance (NMR)

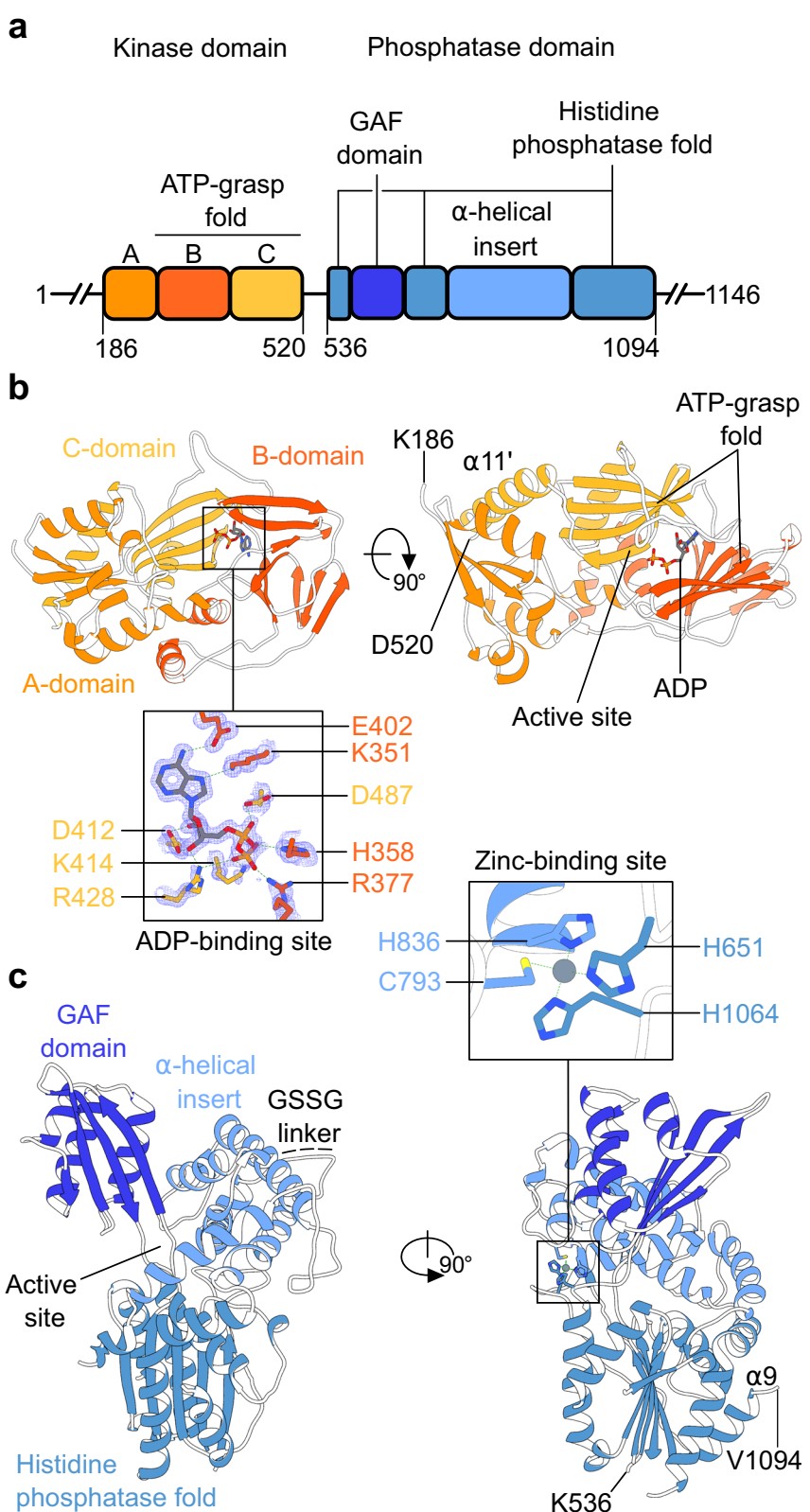

**Fig. 1 | PPIP5K enzymes contain an unusual histidine acid phosphate domain.**
**a** Schematic diagram of the *S. cerevisiae* Vip1 kinase (orange, ScVip1^KD) and phosphatase (blue, ScVip1^PD) domains (https://www.uniprot.org/ Uniprot ID Q06685), highlighting structured regions as blocks (residues 186–520 and 536–1094) and putative linker regions as solid lines (residues 1–185, 521–535, and 1095–1146). **b** Ribbon diagram of the ScVip1^KD structure bound to ADP (in bonds representation) and including an omit 2 ($F_o–F_c$) electron density map

contoured at 1 σ (blue mesh). **c** Ribbon diagram of ScVip1^PD Δ848-918, with the histidine acid phosphatase core and α-helical insertion domain shown in light blue, and the PPIP5K-specific GAF domain in dark blue, respectively. The position of the engineered GSSG linker is indicated with a dotted line. The inlet provides a view of the zinc binding site, with the Zn^2+ ion shown as a sphere (in gray) and the coordinating histidine and cysteine residues depicted in bonds representation.

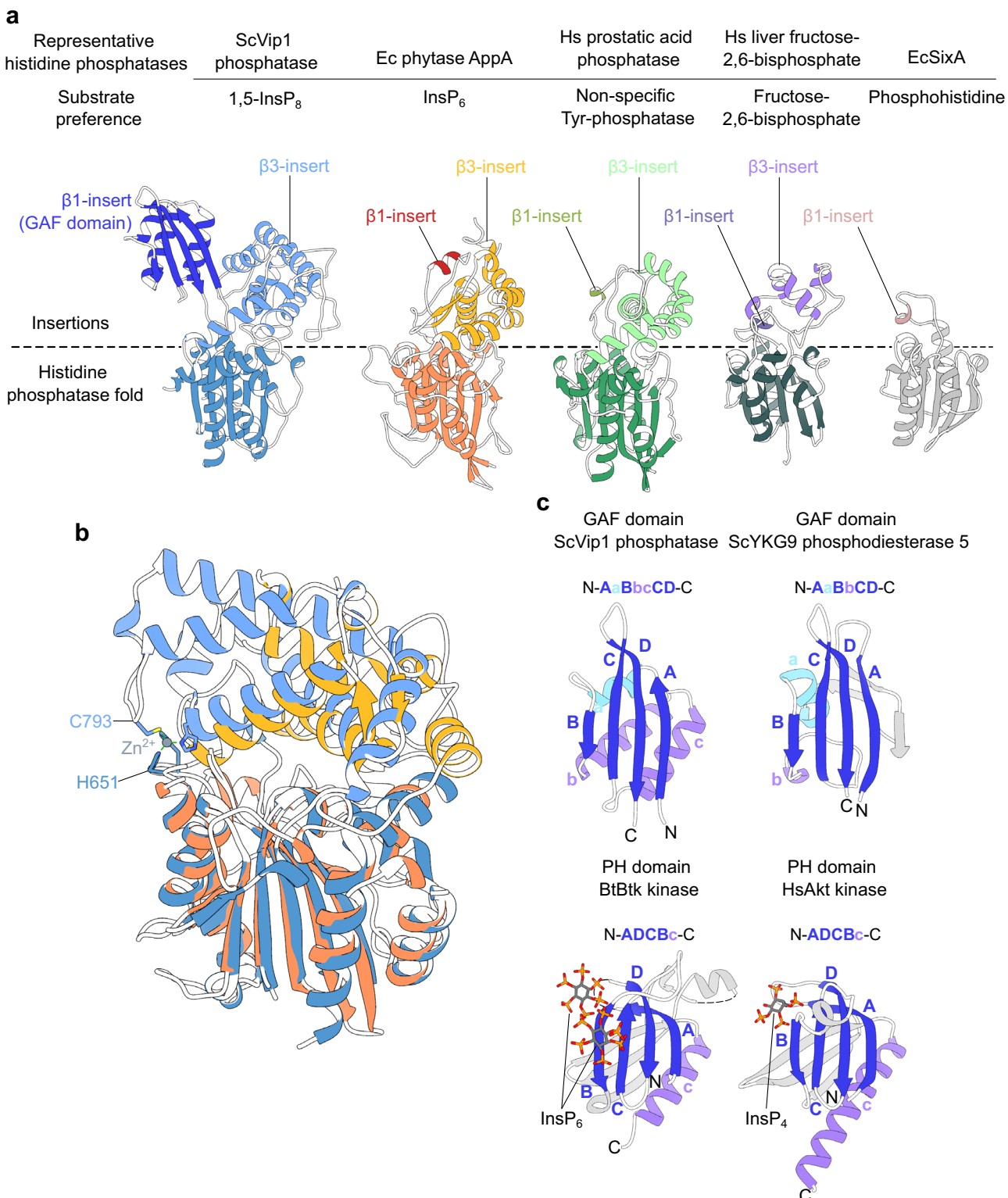

**Fig. 2 | A unique GAF domain in PPIP5Ks replaces the canonical β1-insert in histidine acid phosphatases. a** Structural comparison of ScVip1$^{PD\ \Delta848-918}$ (ribbon diagram, in blue) with the *E. coli* phytase AppA (https://doi.org/10.2210/pdb7z2s/pdb[60], yellow to red), human prostatic acid phosphatase (https://doi.org/10.2210/pdb1nd6/pdb[99], light- to dark-green), human liver fructose-2,6-bisphosphatase (https://doi.org/10.2210/pdb1k6m/pdb[71], purple and dark gray) and the sensor histidine kinase phosphatase SixA from *E. coli* (https://doi.org/10.2210/pdb1ujc/pdb[100], pink and light gray). **b** Structural superposition of ScVip1$^{PD\ \Delta848-918}$ and EcAppA (colors

as in panel A, root mean square deviation r.m.s.d. is -1.1 Å comparing 105 corresponding Cα atoms from the histidine acid phosphatase core). The Zn$^{2+}$ binding site in ScVip1$^{PD}$ is not conserved among other histidine acid phosphatases. **c** Structural comparison of the GAF domain in ScVip1$^{PD}$ and structurally related GAF (https://doi.org/10.2210/pdb1f5m/pdb[53], r.m.s.d. is -0.9 Å comparing 26 corresponding Cα atoms in the β-sheet) and inositol polyphosphate-binding PH domains (https://doi.org/10.2210/pdb4y94/pdb[56], https://doi.org/10.2210/pdb1unq/pdb[55]). Note that GAF and PH domains share similar sized β-sheets with different topologies.

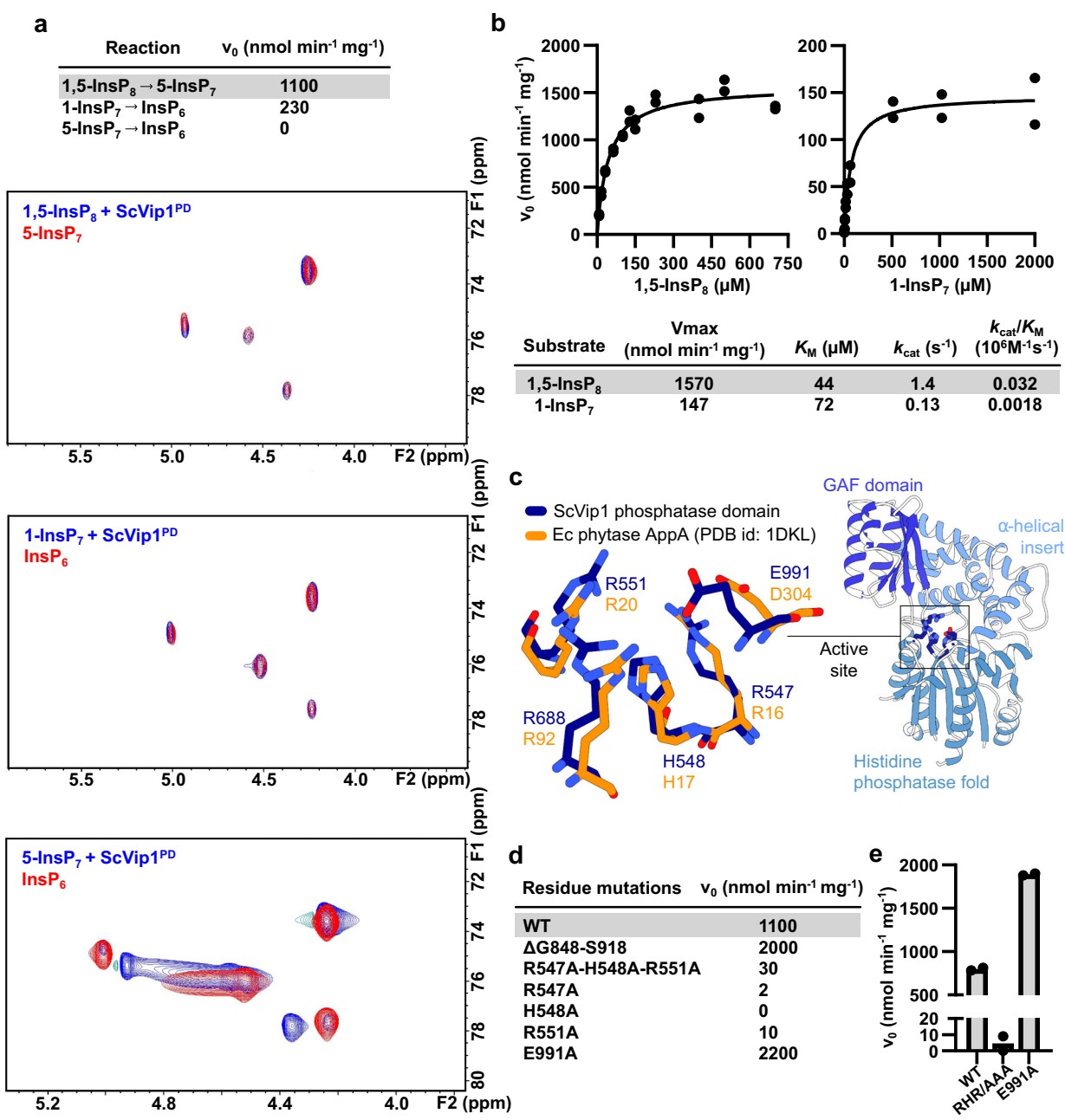

**Fig. 3 | ScVip1$^{PD}$ specifically hydrolyzes 1,5-InsP$_8$ using a conserved phytase active site. a** Table summary of the initial enzyme velocities (v$_0$) for ScVip1$^{PD}$ versus 1,5-InsP$_8$, 1-InsP$_7$, or 5-InsP$_7$ derived from an NMR-based phosphatase assay. Pseudo-2D NMR spectra of ScVip1$^{PD}$ incubated with different substrates (blue traces) are shown below, overlaid with the respective reaction products 5-InsP$_7$, or InsP$_6$ (red traces). **b** Initial enzyme velocities (v$_0$) for ScVip1$^{PD}$–catalyzed hydrolysis of 1,5-InsP$_8$ and 5-InsP$_7$ as a function of inositol pyrophosphate concentration and using a malachite green-based phosphatase assay. The Michaelis–Menten equation was fitted to the data using non-linear regression. Extracted kinetic parameters are summarized in the table as means (two biological replicates were performed, $n$ = 2). **c** Structural superposition of the active sites of ScVip1$^{PD \Delta 848-918}$ (in bond representation, in dark blue) and EcAppA (in orange, https://doi.org/10.2210/pdb1dkl/pdb, r.m.s.d. is -1.0 Å comparing 67 corresponding C$_\alpha$ atoms from the histidine acid phosphatase core). In EcAppA, His17 is the catalytic residue, Asp304 is the proton-donor, and Arg16, Arg20, and Arg92 are involved in InsP$_6$ substrate binding. Initial velocities of 1,5-InsP$_8$ hydrolysis catalyzed by wild-type and mutant versions of ScVip1$^{PD}$, measured by NMR-based (**d**) or malachite green-based (**e**) phosphatase assays ($n$ = 2).

phosphatase assays using $^{13}$C-labeled substrates (Fig. 3a; Supplementary Fig. 5). ScVip1$^{PD}$ is a specific inositol pyrophosphate phosphatase, with a high preference for the 1-position of PP-InsPs (Fig. 3a; Supplementary Fig. 5), as previously reported[25]. In malachite green-based phosphatase assays, we derived similar $K_M$ values for 1,5-InsP$_8$ and 1-InsP$_7$, but the catalytic efficiency of ScVip1$^{PD}$ for 1,5-InsP$_8$ is -18fold higher when compared to 1-InsP$_7$ (Fig. 3b). Notably, 1,5-InsP$_8$ is the reaction product of the N-terminal kinase domain of ScVip1[24].

A sequence motif central to catalysis in histidine acid phosphatases is located in the active site cleft of ScVip1$^{PD}$ sandwiched between the phosphatase core and the α-helical insertion domain (Fig. 3c). The motif with the consensus sequence Arg-His-x-x-Arg (RHxxR, where x can be any amino acid) is structurally conserved between ScVip1$^{PD}$ and bacterial phytases (Fig. 3c). Mutation of either Arg547, His548 or Arg551 to alanine greatly reduces 1,5-InsP$_8$ hydrolysis in NMR- and malachite green-based enzyme assays, respectively (Fig. 3d, e; Supplementary Fig. 5). This is consistent with the established function of

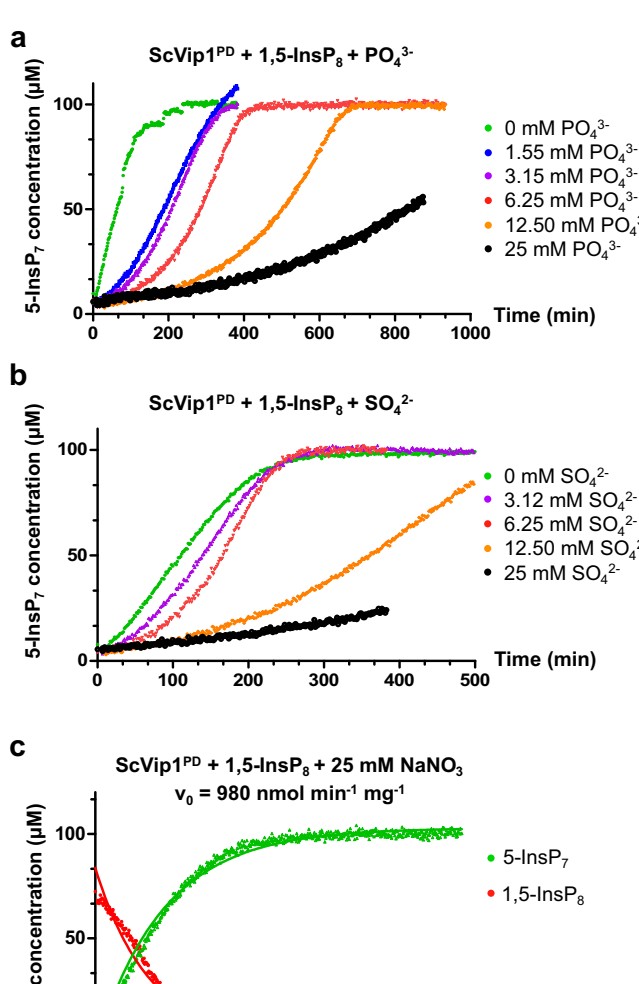

**Fig. 4 | ScVip1$^{PD}$ phosphatase activity is inhibited by inorganic phosphate and sulfate.** NMR time course experiments for ScVip1$^{PD}$-catalyzed hydrolysis of 100 μM 1,5-InsP$_8$ to 5-InsP$_7$ in the presence of increasing concentrations of (**a**) PO$_4^{3-}$, (**b**) SO$_4^{2-}$, or (**c**) at a constant concentration of 25 mM NaNO$_3$. **c** InsP = 1,5-InsP$_8$ (red line) or 5-InsP$_7$ (green line). The measured initial velocity of 980 nmol min$^{-1}$ mg$^{-1}$ is similar to the initial velocity in absence of NaNO$_3$ (1100 nmol min$^{-1}$ mg$^{-1}$, compare Fig. 3a).

the two arginine residues in InsP$_6$ substrate binding in bacterial phytases[57], and the essential role of the nucleophilic acceptor histidine in catalysis[48]. Asp304 in *E. coli* phytase is the proton donor involved in the degradation of the phosphohistidine intermediate[58]. In our ScVip1$^{PD}$ structures, a glutamate residue (Glu991) occupies the position of the phytase proton donor aspartate (Fig. 3c). Mutation of Glu911 to alanine activates the enzyme (Fig. 3d, e; Supplementary Fig. 5, see below). Similarly, deletion of the large disordered loop in the α-helical insertion domain (ScVip1$^{PDΔ848-918}$) moderately activates the engineered enzyme in vitro (Fig. 3d).

In conclusion, the isolated phosphatase domain of ScVip1 is an inositol 1-pyrophosphate phosphatase with a substrate preference for 1,5-InsP$_8$ and with an active site similar to InsP$_6$ metabolizing phytases.

## Inhibition of ScVip1$^{PD}$ by Pi

Gu et al. previously reported that low millimolar concentrations of Pi inhibited the inositol 1-pyrophosphate phosphatase activity of human PPIP5K1 and PPIP5K2[16]. Addition of 10 mM Pi inhibited the phosphatase activity of ScVip1[10]. We found that ScVip1$^{PD}$ is inhibited by Pi in a

concentration-dependent manner, as inferred from NMR-based assays (Fig. 4a). As previously reported for SpAsp1[40], sulfate was less effective than Pi in inhibiting the ScVip1$^{PD}$ phosphatase activity (Fig. 4b). Nitrate had no detectable effect on the hydrolysis of 1,5-InsP$_8$ (Fig. 4c). We conclude that Pi at concentrations found in the yeast cytosol[59], directly inhibits the enzymatic activity of the isolated ScVip1 phosphatase domain.

## GAF domain movements control enzyme activity

All four molecules in the asymmetric unit of our ScVip1$^{PD}$ crystals display the GAF domain in an "open" conformation (Fig. 5a; Supplementary Fig. 6; Supplementary Table 1). In ScVip1$^{PD Δ848-918}$ crystals, two molecules in the asymmetric unit also adopt the "open" conformation (Fig. 5a; Supplementary Fig. 6; Supplementary Table 1). Crystal packing interactions between two adjacent GAF domains in our ScVip1$^{PD Δ848-918}$ crystals are mediated in part by a second Zn$^{2+}$ binding site. The Zn$^{2+}$ ion is coordinated along a pseudo two-fold axis involving residues His573 and Glu575 from the GAF domain (Supplementary Fig. 6). Given that ScVip1$^{PD}$ behaves as a monomer in solution (Supplementary Fig. 7) and that the histidine and glutamate residues are not conserved among other PPIP5Ks (Supplementary Fig. 2), the second Zn$^{2+}$ coordination site is likely a crystallization artifact (Supplementary Fig. 6).

Next, we replaced the catalytic loop residues Arg547, His548 and Arg551 in ScVip1$^{PD Δ848-918}$ with alanine and obtained crystals in complex with the substrate 1,5-InsP$_8$ diffracting at ~2.4 Å resolution (ScVip1$^{PD Δ848-918 RHR-AAA}$, Supplementary Table 1). Notably, the GAF domain adopts a "closed" conformation in the substrate-bound structure (Fig. 5a). We hypothesized that the movements of the GAF domain are enabled by a hinge region in close proximity to the catalytic loop (RHxxR motif) of the enzyme (Fig. 5a, b). Consequently, mutation of the hinge residues Pro553, Gly646 and Gly647 (P553V-G646V-G647A) inactivated ScVip1$^{PD}$ in NMR- and malachite green-based phosphatase assays, respectively (Fig. 5b–d). This suggests that movement of the GAF domain is part of the enzymatic cycle of ScVip1$^{PD}$. Notably, the catalytic loop (Arg547, His548, Arg551) does not appear to be involved in the domain movement when comparing the open and closed conformation structures (Fig. 5a). The conserved Zn$^{2+}$ binding site in ScVip1$^{PD}$ positions His836 to form a salt bridge with Asp550, apparently stabilizing the catalytic loop (Fig. 5b). Mutation of the Zn$^{2+}$ binding site residues His651 or Cys793 to alanine reduces the phosphatase activity in our enzyme assays, suggesting that the Zn$^{2+}$ binding site is involved in the structural stabilization of the enzyme and, importantly, of its catalytic loop (Fig. 5b–e). Consistent with this, mutation of Zn$^{2+}$ coordinating residues reduces the structural stability of ScVip1$^{PD}$, whereas excess of ZnCl$_2$ further stabilizes the enzyme in thermal shift assays in vitro (Fig. 5e, Supplementary Fig. 7).

## The GAF domain is involved in substrate binding and channeling

In our ScVip1$^{PD Δ848-918 RHR-AAA}$–1,5-InsP$_8$ complex structure, we located a crystallographic dimer stabilized by an intermolecular disulfide bridge involving a cysteine (Cys793) that, in apo structures, is part of the zinc-binding site (Supplementary Fig. 8). One molecule of 1,5-InsP$_8$ is bound to each monomer, either on the surface of the enzyme involving the GAF domain or near the catalytic center (Fig. 6a; Supplementary Fig. 8). The GAF domain adopts a closed conformation in both monomers (Fig. 6a). In both binding sites, all phosphate groups of 1,5-InsP$_8$ are well defined by electron density (Fig. 6b), with the 1-β-phosphate facing away from the active site (Fig. 6c). Conserved basic residues from the GAF domain and the α-insertion domain contribute to the formation of the outer substrate binding surface (Fig. 6c). A large network of lysine residues from the GAF domain and from the base of the α-insertion domain forms the second substrate binding surface, which is located in close proximity to the RHxxR catalytic loop (Fig. 6c). Most of the basic residues contributing to the

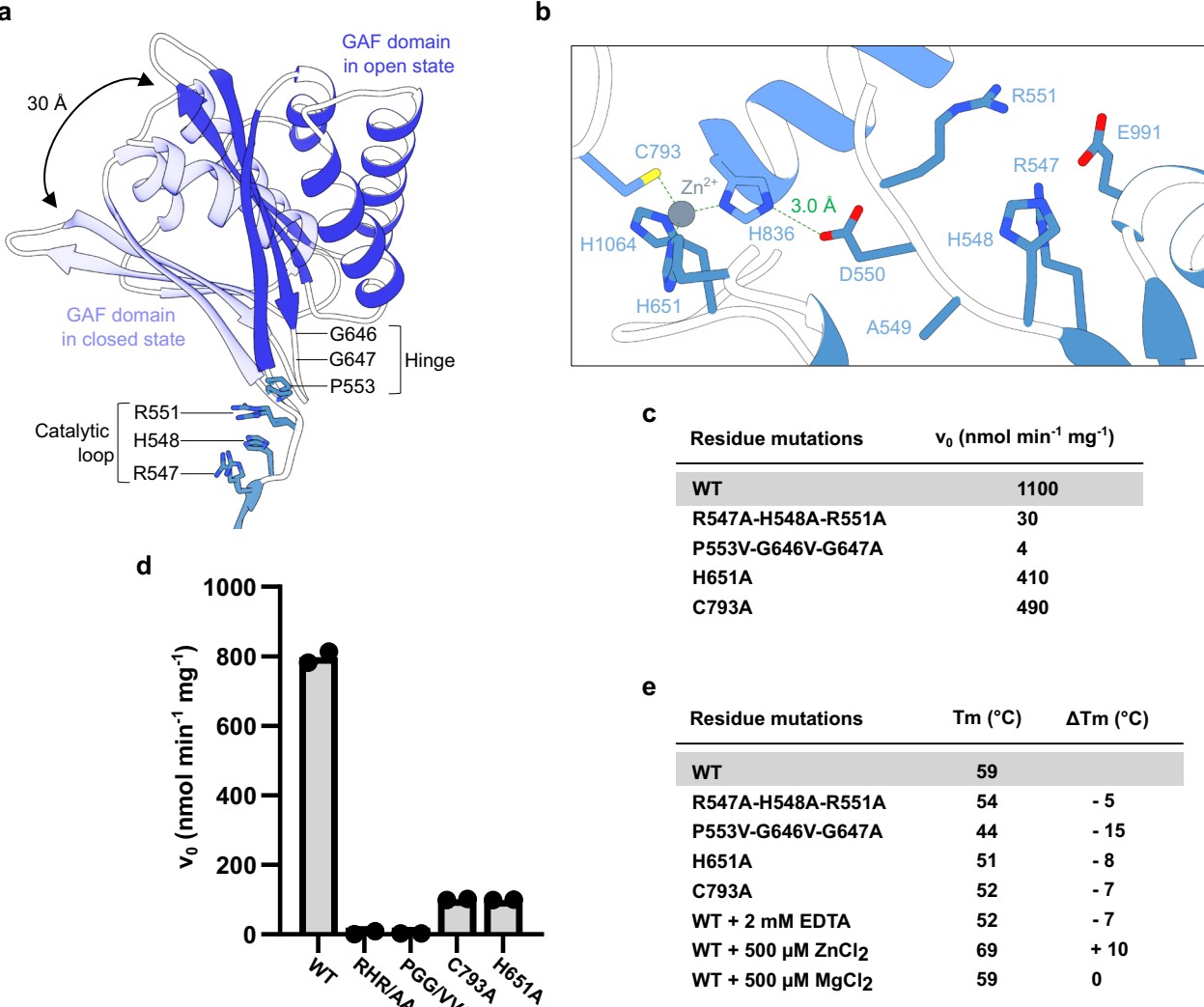

**Fig. 5 | ScVip1$^{PD}$ is regulated by GAF domain movements and stabilized by a zinc-binding site. a** Structural superposition of apo ScVip1$^{PD \Delta 848-918}$ reveals the GAF insertion domain in either its open (chain A, dark blue ribbon diagram) or closed (chain D, in light blue) conformation. The RHxxR and hinge regions are shown alongside (in bonds representation) **b** Details of the ScVip1$^{PD \Delta 848-918}$ active site: Asp550 stabilizes the catalytic loop by engaging in a salt bridge (green dotted line) with the Zn$^{2+}$ ion- coordinating His836. The active site and Zn$^{2+}$ coordinating residues are depicted in bonds representation, the Zn$^{2+}$ ion is shown as a gray sphere. Initial velocities of 1,5-InsP$_8$ hydrolysis catalyzed by wild-type or mutant version of ScVip1$^{PD}$, measured by NMR (**c**) or malachite green (**d**) phosphatase assays ($n = 2$). **e** Table summarizing the thermal melting profiles of wild-type or mutant ScVip1$^{PD}$. The ΔTm (in °C) represent the difference between the melting temperature of a designated ScVip1$^{PD}$ mutant protein and the wildtype control.

outer and the inner 1,5-InsP$_8$ binding surfaces map to the central β-sheet of the GAF domain (Fig. 6c, d). Comparison of the outer and inner substrate binding sites in ScVip1$^{PD}$ suggests that the 1,5-InsP$_8$ substrate may undergo a -90° rotation upon entering the active site (Fig. 6d).

We next compared the open conformation of apo ScVip1$^{PD}$ with the substrate-bound structures (Fig. 7a). We found that the conformational changes strongly alter the surface charge distribution of the enzyme, with the GAF domain opening and closing on a large and well conserved basic surface area surrounding the substrate binding sites and formed by the GAF and α-insertion domains (Fig. 7a, b). By mapping the positions of the 1,5-InsP$_8$ molecules bound to the outer and inner substrate binding sites, we uncovered a potential substrate binding channel (Fig. 7c). We hypothesize that the 1,5-InsP$_8$ substrate first targets the outer substrate binding site of ScVip1$^{PD}$ in its open conformation. Substrate binding to this outer site may trigger closure of the GAF domain, resulting in rotation of the substrate and its translocation to the active site (Fig. 7c). In its

closed conformation, the PPIP5K GAF domain contributes to the formation of a substrate channel that may facilitate transport of 1,5-InsP$_8$ to the catalytic center as well as the exit of the 5-InsP$_7$ product from the enzyme (Fig. 7c).

Mutation of highly conserved residues from the outer substrate binding site (Lys558, Lys605, Lys732, Lys817 to alanine) (Fig. 7d) inactivated the enzyme (Fig. 7e, f), suggesting that 1,5-InsP$_8$ is indeed initially bound to the surface of the enzyme before entering the active site. We next mutated residues that contribute to the PPIP5K substrate channel. Mutation of Lys817, Asp819, and Ser821 originating from the α-helical insertion domain (Fig. 7d) moderately reduced the 1,5-InsP$_8$ phosphohydrolase activity of ScVip1$^{PD}$ in both NMR- and malachite green-based assays (Fig. 7e, f). In contrast, mutation of 1,5-InsP$_8$ binding residues from the GAF domain (Glu576, Lys623, Gln625, Lys627) inactivated the enzyme (Fig. 7d–f). Mutations close to the active site of the enzyme (Lys554, Lys557, Lys644) likewise reduced the activity of ScVip1$^{PD}$ (Fig. 7d–f), but in addition structurally destabilized the enzyme (Supplementary Fig. 7). We therefore generated an additional

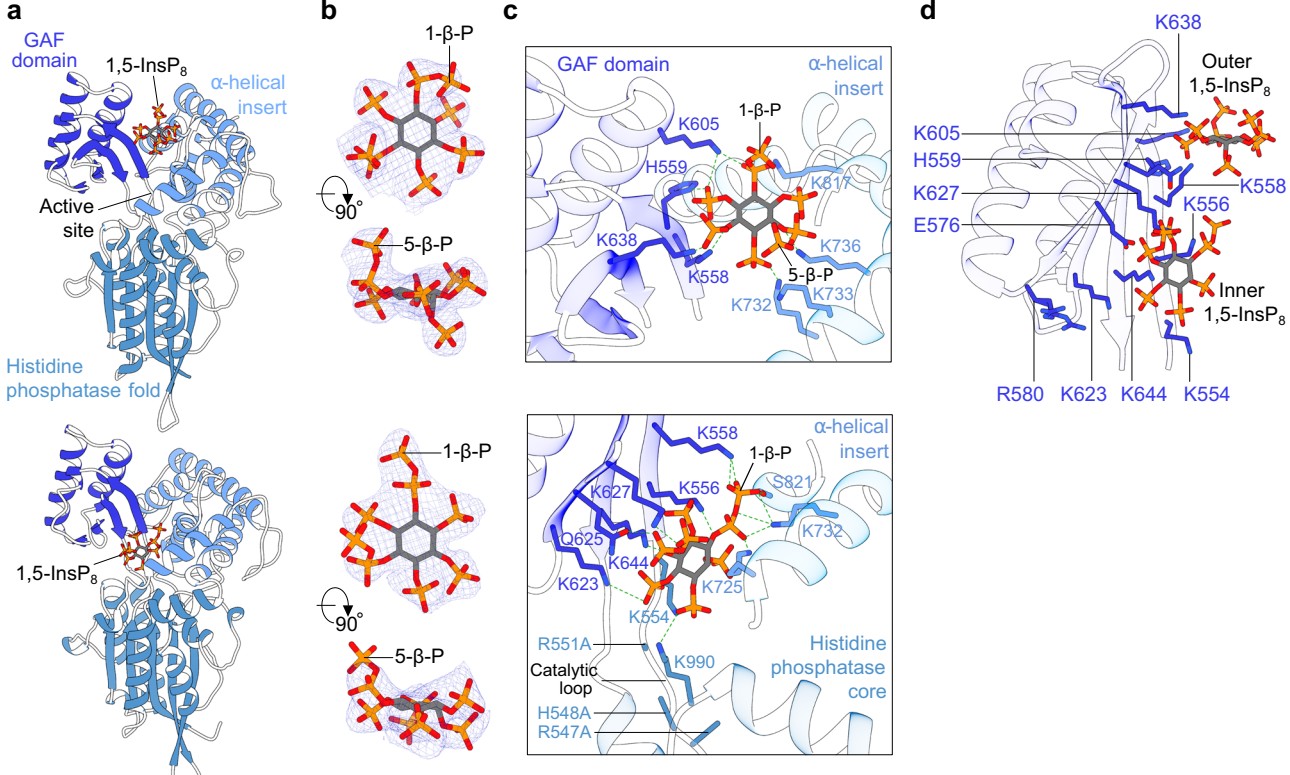

**Fig. 6 | The GAF domain of ScVip1$^{PD}$ binds 1,5-InsP$_8$ in two capture sites.**
**a** Ribbon diagrams of ScVip1$^{PD\,\Delta848-918\,RHR-AAA}$ bound to 1,5-InsP$_8$ (in bonds representation). 1,5-InsP$_8$ binds either at the surface of ScVip1$^{PD\,\Delta848-918\,RHR-AAA}$ (chain A, top) or close to the active site (chain B, bottom). **b** Detailed view of the 1,5-InsP$_8$ molecules shown in two orientations and including a 2 ($F_o$–$F_c$) electron density map (blue mesh) contoured at 2.5 σ in chain A (top) or at 1 σ in chain B (bottom), respectively. "1-β-P" and "5-β-P" designate the positions of the β-phosphate of the pyrophosphate group attached to the position 1 or 5 of the *myo*-inositol ring.
**c** Detailed view of the 1,5-InsP$_8$ binding sites. The residues interacting with 1,5-InsP$_8$ are shown in bonds representation, hydrogen bonds are indicted by dotted lines (in green). **d** Ribbon diagram of the isolated GAF domain from the ScVip1$^{PD\,\Delta848-918\,RHR-AAA}$ structure, depicting the relative positions of the outer (chain A) and inner (chain B, modeled) 1,5-InsP$_8$ substrate (in bonds representation) binding sites.

single Lys554 to alanine mutant, which is structurally intact (Supplementary Fig. 7) but which also showed very low enzyme activity in our assays (Fig. 7d–f). Taken together, our structure-function analyses suggest that PPIP5K phosphatase domains contain two 1,5-InsP$_8$ binding sites that contribute to substrate channeling into the active site.

## An active site glutamate mediates substrate specificity of ScVip1$^{PD}$

As described above, the canonical phytase proton donor aspartate is replaced by Glu991 in ScVip1, which adopts different conformations in our apo and substrate-bound crystal structures (Figs. 3c; 8a, b). Replacement of Glu991 with alanine activates ScVip1$^{PD}$ (Fig. 3d, e). In addition, we observed that the ScVip1$^{PD}$ Glu991 to Ala mutant readily accepts 1-InsP$_7$ as a substrate in both NMR and malachite green-based phosphatase assays, but remains unable to hydrolyze InsP$_6$ (Fig. 8c, d). It has been previously reported that mutation of Asp304 in AppA strongly alters the stereospecificity of the *E. coli* enzyme[60]. Notably, multiple inositol polyphosphate phosphatases that lack strong stereospecificity for InsP$_6$ are characterized by an alanine residue at this position (Fig. 8a, b)[61]. We conclude that Glu991, which is conserved among PPIP5Ks from different species (Supplementary Fig. 2) is involved in the selection of the 1,5-InsP$_8$ substrate. The inositol ring occupies a different position in our ScVip1$^{PD}$–1,5-InsP$_8$ complex structure when compared to known phytases (Fig. 8b). We speculate that this is an adaptation to the hydrolysis of a pyrophosphate substrate. The long and uniquely shaped substrate channel may play an additional role in the selection of 1,5-InsP$_8$ (Fig. 7c).

## Mutation of key motifs inhibits the function of the VIH2 phosphatase domain in planta

We next sought to functionally characterize the substrate binding and catalytic mechanisms of ScVip1$^{PD}$ in yeast. Previous studies in fission yeast revealed a requirement of the PPIP5K Asp1 in ensuring fidelity of chromosome segregation by increasing microtubule stability[35]. Like the fission yeast ortholog, ScVip1 is encoded by a non-essential gene in budding yeast[24]. We analyzed the Vip1 genetic interaction network in numerous Synthetic Genetic Array (SGA) screens to identify mutant backgrounds in which Vip1 function becomes critical[62,63]. These studies revealed synthetic sick/lethal interactions between Vip1 and components required to maintain chromosome stability[62]. One such component is Bim1[64], a conserved +end microtubule binding protein, which in complex with Kar9 forms the cortical microtubule capture site and delays the exit from mitosis when the mitotic spindle is oriented abnormally[65]. We generated a vip1Δbim1Δ mutant strain that is synthetic sick (Supplementary Fig. 9a) and used it for complementation assays. We found that both expression of wild-type full-length ScVip1 and of a mutant targeting the RHxxR motif (ScVip1$^{R547A-H548A-551A}$), which impairs the catalytic function of the phosphatase (Fig. 3d, e), fully restored growth to wild-type like levels (Supplementary Fig. 9a). The ScVip1$^{E991A}$ activating mutant (Fig. 3d, e) also behaved similar to wild type, as did a mutant targeting the outer substrate binding site in the phosphatase domain (ScVip1$^{K558A-K605A-K732A}$) (Supplementary Fig. 9a).

In an alternative approach, we found that overexpression of the isolated kinase domain is lethal for yeast cells grown on low phosphate (Pi) containing media (ScVip1$^{KD}$ in Supplementary Fig. 9b). This effect

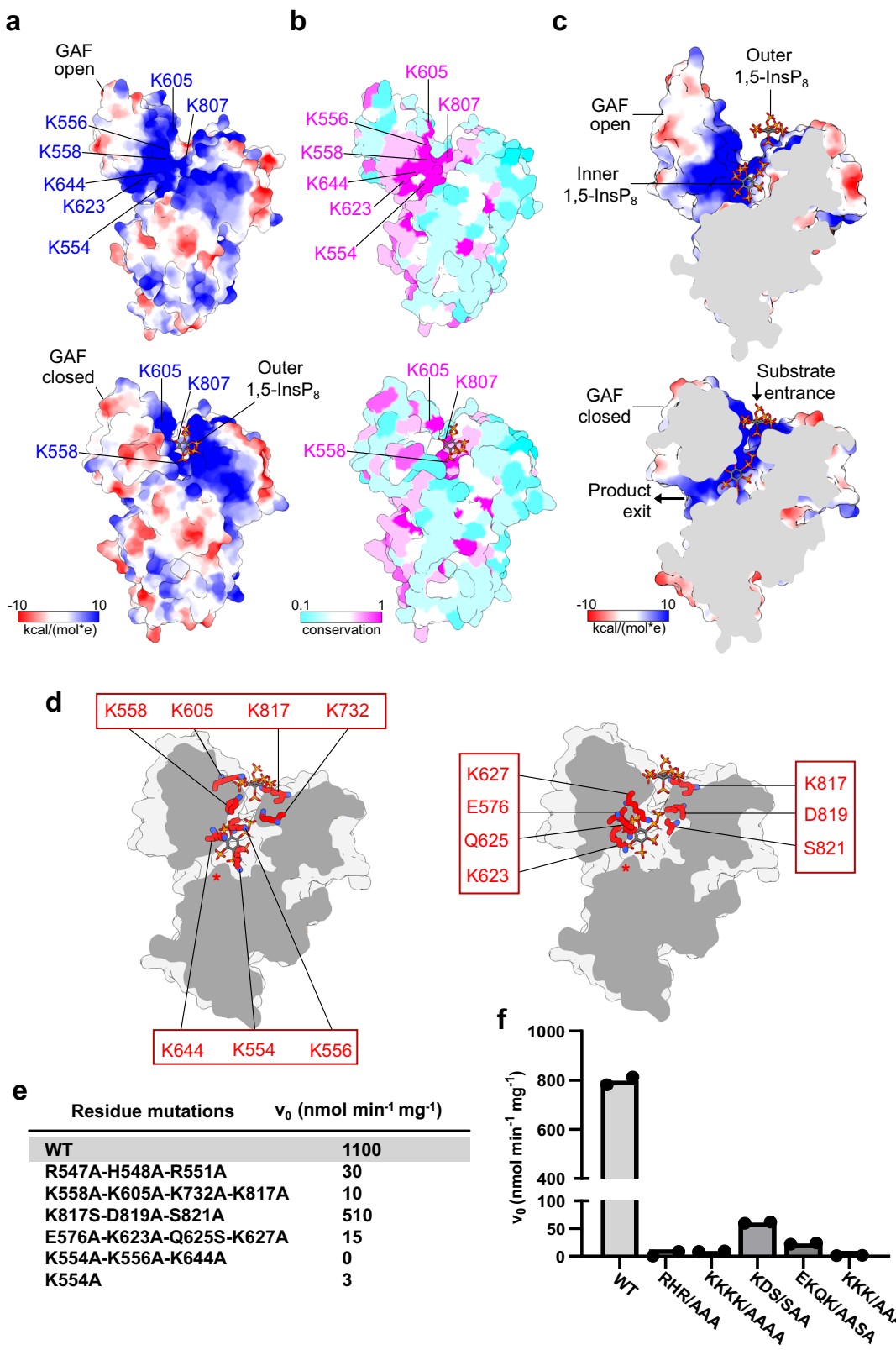

| Residue mutations | $v_0$ (nmol min$^{-1}$ mg$^{-1}$) |
|---|---|
| **WT** | **1100** |
| R547A-H548A-R551A | 30 |
| K558A-K605A-K732A-K817A | 10 |
| K817S-D819A-S821A | 510 |
| E576A-K623A-Q625S-K627A | 15 |
| K554A-K556A-K644A | 0 |
| K554A | 3 |

was not observed when overexpressing either full-length or kinase-dead (ScVip1$^{KD\ K414A\cdot D487A}$) versions of the enzyme (Supplementary Fig. 9b). In contrast, overexpression of either wild-type or catalytically inactive versions of ScVip1$^{PD}$ was not lethal when yeast were grown on low Pi-containing media (Supplementary Fig. 9b).

Returning to our original plan to characterize the catalytic mechanism of plant PPIP5K phosphatases (see above), we next mapped all residues and sequence motifs characterized in ScVip1$^{PD}$ to AtVIH2 from *Arabidopsis thaliana*, which shares 25% sequence identity with ScVip1$^{PD}$ (Fig. 9a; Supplementary Fig. 2). We have previously reported that constitutive expression of a phosphatase-dead variant of full-length AtVIH2 (AtVIH2$^{R372A\cdot H373A}$) in Col-0 wild-type background reduced rosette size and intracellular Pi levels[10]. We used these phenotypes to perform structure-function studies with the AtVIH2

**Fig. 7 | A GAF domain-induced substrate channel transports 1,5-InsP₈ to the active site. a** Electrostatic surface charge distribution of the apo ScVip1$^{PD\ \Delta848-918}$ (top) and 1,5-InsP₈-bound ScVip1$^{PD\ \Delta848-918\ RHR-AAA}$ (bottom) structures (in surface representation). **b** Evolutionary conservation (from cyan, variable to pink, invariant) of surface residues mapped onto the apo ScVip1$^{PD\ \Delta848-918}$ (top) and 1,5-InsP₈-bound ScVip1$^{PD\ \Delta848-918\ RHR-AAA}$ (bottom) structures. Views as in (**a**). Residue conservation was derived from the multiple sequence alignment shown in Supplementary Fig. 2 and analyzed with Consurf[101]. **c** Cutaway front view of the apo ScVip1$^{PD\ \Delta848-918}$ (shown as

molecular surface), with the outer and inner 1,5-InsP₈ molecules depicted in bonds representation (top). Same view for the 1,5-InsP₈-bound ScVip1$^{PD\ \Delta848-918\ RHR-AAA}$ structure (bottom). The closed GAF domain forms the substrate channel. **d** Groups of mutated residues involved in 1,5-InsP₈ channeling (in bonds representation) mapped onto the ScVip1$^{PD\ \Delta848-918\ RHR-AAA}$ structure (shown as molecular surface). The active site of ScVip1$^{PD}$ is indicated with a red asterisk. Initial velocities of 1,5-InsP₈ hydrolysis for wild-type or mutant versions of ScVip1$^{PD}$, measured by NMR-based (**e**) or malachite green-based (**f**) phosphatase assays ($n = 2$).

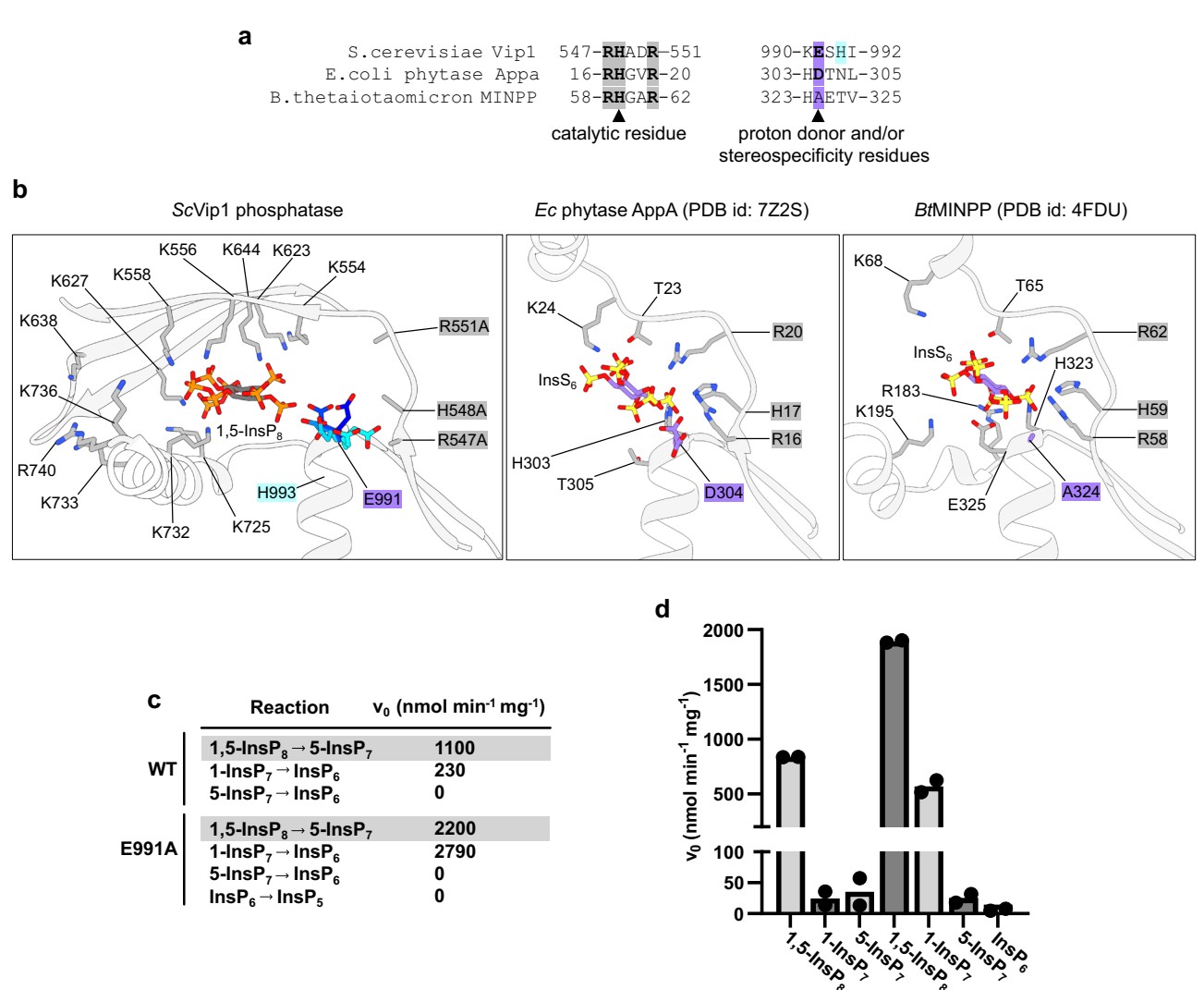

**Fig. 8 | A conserved glutamate controls substrate sterospecificity in ScVip1$^{PD}$.**
**a** Structure-based sequence alignment of the presumed catalytic motifs in ScVip1$^{PD}$, *E. coli* AppA, and the multiple inositol polyphosphate phosphatase MINPP from *Bacteroides thetaiotaomicron*. The RHxxR motif is conserved among all enzymes, while the proton donor aspartate in EcAppA is replaced by Glu991 in ScVip1$^{PD}$ and by an alanine residue in MINPP. **b** Structural comparison of the ScVip1$^{PD}$, EcAppA (https://doi.org/10.2210/pdb7z2s/pdb[60]) and MINPP (https://

doi.org/10.2210/pdb4fdu/pdb[61]) active sites. Glu991 (in bonds representation, colored from cyan to blue) adopts different conformation in our structures. Asp304 controlling substrate stereospecificity in EcAppA[60], and Ala324 in the highly promiscuous MINPP enzyme are highlighted in blue. Initial velocities of 1,5-InsP₈, 1-InsP₇, and 5-InsP₇ hydrolysis for wild-type or Glu991Ala versions of ScVip1$^{PD}$, measured by NMR-based (**c**) or malachite green-based (**d**) phosphatase assays ($n = 2$).

phosphatase domain. Constitutive expression of wild-type AtVIH2 had no detectable effect on either rosette area or cellular Pi levels (Fig. 9b–e). Expression of a kinase-dead version of the enzyme (AtVIH2$^{K219A-D292A}$) resulted in hyperaccumulation of cellular Pi, as previously reported (Fig. 9b–e)[10]. Three different mutant combinations targeting the catalytic core of the phosphatase domain in Arabidopsis VIH2 (compare Fig. 3c–e) were all associated with significantly reduced

rosette areas and cellular Pi levels (highlighted in blue in Fig. 9a–e). Mutations targeting the conserved structural Zn$^{2+}$ binding site (compare Fig. 5b–e) were also associated with reduced rosette areas and cellular Pi levels (highlighted in purple in Fig. 9a–e). A mutant in the phosphatase hinge region showed less severe growth phenotypes and only slightly reduced Pi levels (shown in red in Fig. 9a–e), whereas the corresponding mutations in ScVip1$^{PD}$ strongly affected the activity and

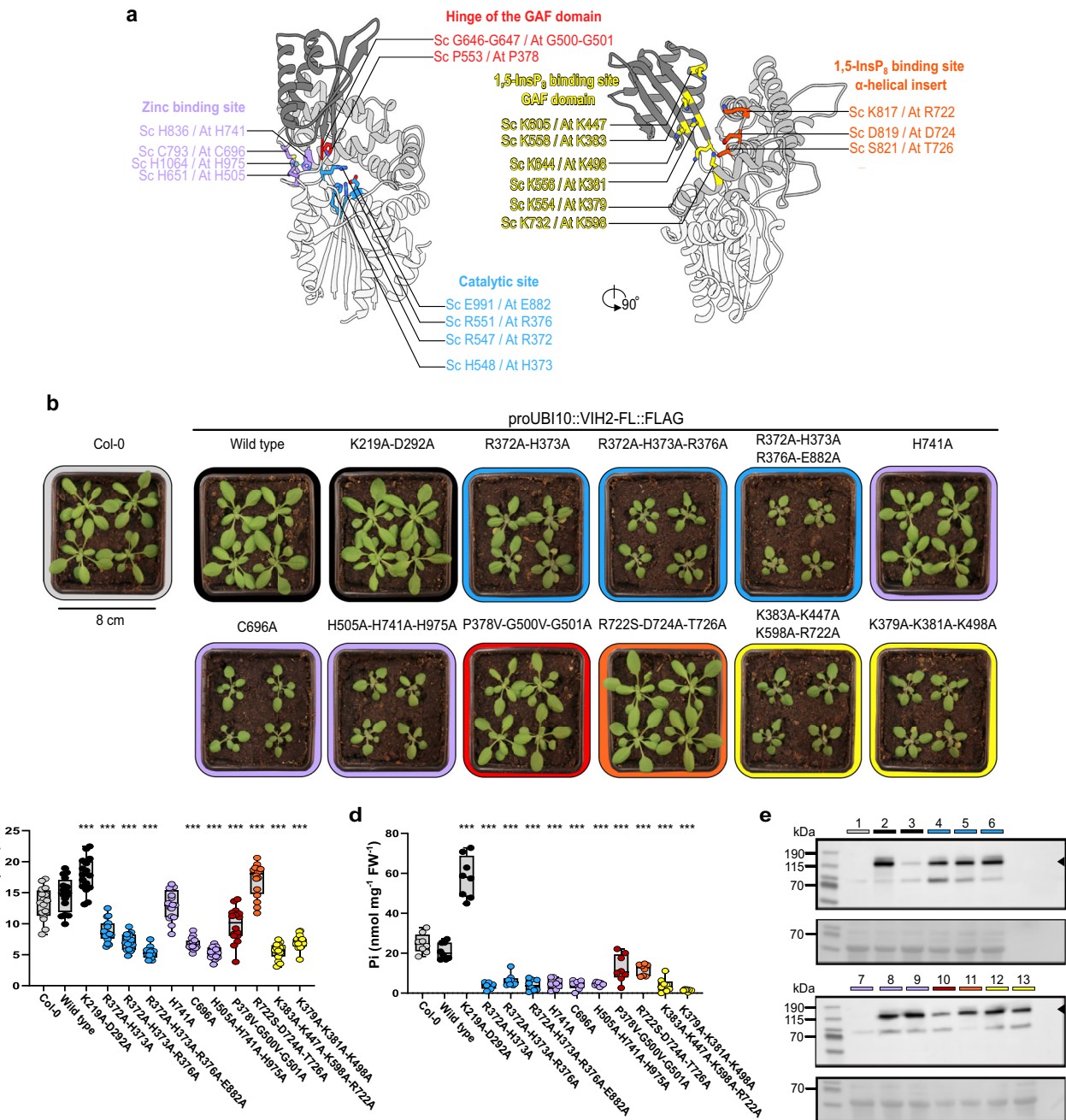

**Fig. 9 | Structure-function analysis of the AtVIH2 phosphatase domain.**
**a** Ribbon diagram of the ScVip1$^{PD \Delta848-918}$ structure, with the histidine phosphatase core in white, the α-helical insert in light gray, and the GAF domain in dark gray. Residues involved in phosphatase catalysis, $Zn^{2+}$ ion coordination, GAF domain movement, and 1,5-InsP$_8$ binding are shown in bond representation. Corresponding residues in AtVIH2 are also indicated, illustrating the conserved functional regions. **b** Rosette phenotype of 3 weeks old Col-0 wild-type plants and lines expressing full-length (FL) Flag-tagged versions of AtVIH2 under the control of the ubiquitin-10 (UBI10) promoter. **c** Quantification of the rosette area. Box plots show 16 plants per genotype (center black line, median; box, interquartile range (IQR); whiskers, lowest/highest data point within 1.5 IQR of the lower/upper quartile; raw data are

denoted by a colored dot). Multiple comparisons of the genotypes vs. Col-0 were performed according to a Dunnett test[102,103] as implemented in GraphPad prism v10.3.0 (***$p < 0.001$, **$p < 0.005$, *$p < 0.01$) ($n = 16$). **d** Quantification of cellular Pi levels. For each genotype, 8 individual plants were measured using 3 technical replicates ($n = 8$). The estimated Pi concentration was normalized by tissue fresh weight. **e** Western blot of Flag-tagged AtVIH2. The theoretical molecular mass of AtVIH2 is ~118 kDa (indicated by a black arrow; 1:Col-0, 2:AtVIH2$^{WT}$, 3:AtVIH2$^{K219A-D292A}$, 4:AtVIH2$^{R372A-H373A}$, 5:AtVIH2$^{R372A-H373A-R376A}$, 6:AtVIH2$^{R372A-H373A-R376A-E882A}$, 7:AtVIH2$^{H741A}$, 8:AtVIH2$^{C696A}$, 9:AtVIH2$^{H505A-H741A-H975A}$, 10:AtVIH2$^{P378V-G500V-G501A}$, 11:AtVIH2$^{R722S-D724A-T726A}$, 12:AtVIH2$^{K383A-K447A-K598A-R722A}$, 13:AtVIH2$^{K379A-K381A-K498A}$. These experiments were performed twice, compare Fig. S10.

structural stability of the enzyme in vitro (Fig. 5a, c–e). Consistent with the moderate effect of mutations in the α-helical insert contributing to the outer 1,5-InsP$_8$ binding site (Fig. 7d–f), the corresponding AtVIH2$^{R722S-D724A-T726A}$ mutant looks similar to the AtVIH2 wild-type expressing control (Fig. 9a–e). Importantly, mutations targeting either

the outer or inner GAF domain binding surfaces (compare Fig. 7d–f) both resulted in a severe reduction in rosette area and in low Pi levels (shown in yellow in Fig. 9a–e). In conclusion, our mutational analysis of the AtVIH2 phosphatase domain confirms the importance of the catalytic center of the enzyme and highlights the critical contributions of

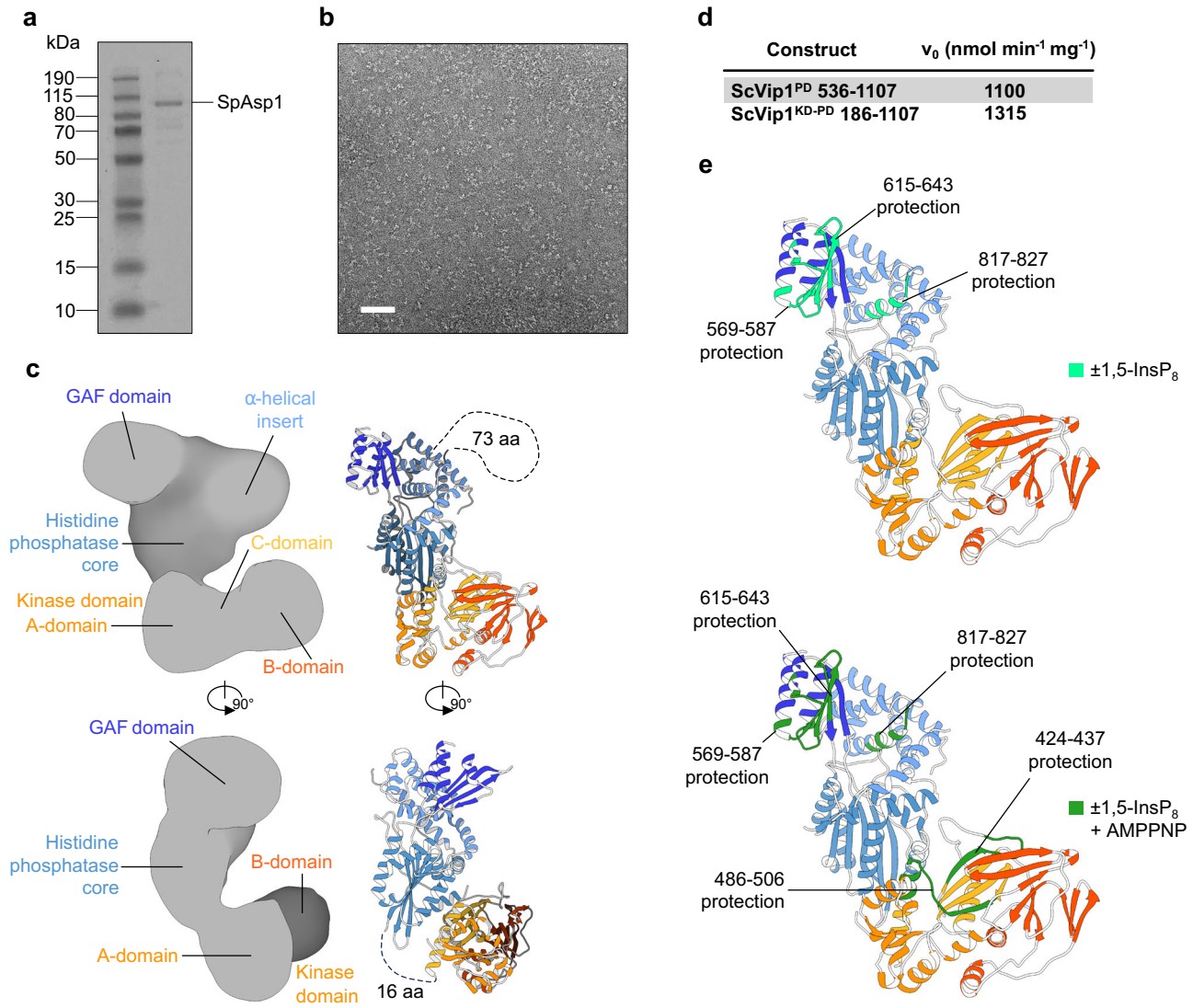

**Fig. 10 | Arrangement of the the kinase and phosphatase domains in a full-length PPIP5K. a** Coomassie blue-stained SDS-PAGE of purified SpAsp1. **b** A representative micrograph of the negatively stained SpAsp1 used for image processing. 1565 micrographs were collected in total. The scale bar corresponds to 50 nm. **c** Low resolution envelope of SpAsp1 determined by negative stain electron microscopy (left). Individual crystal structures of ScVip1^KD (orange ribbon diagram) and ScVip1^PD Δ848-918 (blue ribbon diagram) were docked into the L-shaped negative stain map of SpAsp1 (right). The linkers truncated in the crystal structures of ScVip1 are potentially visible in the SpAsp1 envelope (black dashed lines). **d** Initial velocities of 1,5-InsP$_8$ hydrolysis catalyzed by ScVip1^PD or ScVip1^KD-PD, measured by NMR-based phosphatase assays. **e** Hydrogen-deuterium exchange data mapped on the ScVip1^KD-PD model. The protected regions in ScVip1 after incubation with non-hydrolysable 1,5-InsP$_8$ (PCP-IP$_8$) are shown in light green, the protected regions after incubation with non-hydrolysable 1,5-InsP$_8$ (PCP-IP$_8$) and ATP (AMPPNP) analogs, respectively are shown in dark green.

the outer and inner 1,5-InsP$_8$ binding sites in the PPIP5K-specific GAF domain (Fig. 9; Supplementary Fig. 10).

### The ScVip1 kinase and phosphatase domains are independent enzymatic modules

We expressed and purified full-length SpAsp1 from *E. coli* (Fig. 10a), as previously described[40], applied the apo enzyme to carbon-coated grids and performed negative-stain electron microscopy (Fig. 10b, see "Methods"). We calculated a low-resolution envelope from two-dimensional projections of single particles and docked the experimental structures of the sequence-related ScVip1 kinase and phosphatase domains (Fig. 10c, Supplementary Fig. 2). Full-length SpAsp1 adopts a L-shaped conformation (Fig. 10c). The N-terminal kinase and the C-terminal phosphatase domains are perpendicular to each other (Fig. 10c). The exact orientation of the kinase domain is somewhat ambiguous but is reasonably constrained by the size of the 16 amino

acid linker (Fig. 10c). In our model, the catalytic centers of the kinase and phosphatase modules are far apart, at least in the conformation adopted by the apo enzyme on the electron microscope grid (Fig. 10c). Consistent with this more open conformation of PPIP5K, the inositol 1-pyrophosphate phosphatase activity of ScVip1^KD-PD (residues 186–1107) is comparable to that of ScVip1^PD (Fig. 10d). Next, we performed hydrogen/deuterium exchange mass spectrometry experiments on ScVip1^KD-PD. Addition of 1,5-InsP$_8$ to the apo enzyme protects regions from the α-helical insertion domain and the GAF domain (Fig. 10e; Supplementary Fig. 11), further supporting our substrate binding model for ScVip1^PD (Fig. 6c), and the proposed dynamics of the GAF domain (Figs. 5a, 7a–c). Addition of an ATP substrate analog reveals additional protected regions in the vicinity of the active site of the kinase domain (Fig. 10e; Supplementary Fig. 11). No changes consistent with the formation of new kinase – phosphatase domain interfaces are observed upon addition of either the phosphatase or

kinase substrate (Fig. 10e; Supplementary Fig. 11). Taken together, these experiments suggest that the PPIP5K kinase and phosphatase modules are enzymatically and structurally independent entities.

## Discussion

PPIP5Ks play a central role in inositol pyrophosphate metabolism and signaling[2]. The enzyme activity, substrate specificity, 3-dimensional structure and catalytic mechanism of the N-terminal kinase domain of PPIP5K are well established[26–28,40,46,66], but the C-terminal phosphatase domain has only been partially characterized biochemically[25,29,40–42], and has resisted structural analysis[2]. We were able to overcome this problem by focusing on the phosphatase domain of ScVip1, by using a eukaryotic expression system, by fine-mapping the phosphatase domain boundaries and by engineering of a large unstructured loop (see "Methods", Supplementary Table 1). The structure of ScVip1$^{PD}$ confirms its evolutionary relationship with bacterial and animal histidine acid phosphatases[24]. We found that the active site of ScVip1$^{PD}$ is very similar to that of InsP$_6$ hydrolyzing phytases, with both enzymes sharing the structurally conserved RHxxR catalytic motif[24,47] (Fig. 3c–e). It is noteworthy that ScVip1$^{PD}$ specifically catalyzes the hydrolysis of a phosphoanhydride bond, not a phosphomonoester as seen with other histdine acid phosphatases (Fig. 3).

A second catalytic histidine (His993 in ScVip1, His807 in SpAsp1) was originally defined by sequence comparison with *E. coli* AppA[23,24], where the corresponding His303 is involved in InsP$_6$ substrate binding and in catalysis[48,57]. Structural comparisons of EcAppA and ScVip1$^{PD}$ reveal that this motif has been incorrectly assigned. His303 in the bacterial phytase corresponds to a poorly conserved lysine residue (Lys990) in ScVip1 (Supplementary Figs. 11, 2). His993 in ScVip1 or His807 in SpAsp1 are not part of the active site but rather map to an α-helical segment buried in the phosphatase domain (Supplementary Fig. 12). Thus, substitution of SpAsp1 H807 with alanine may disrupt the structural integrity of the enzyme rather than affect its catalytic activity, providing an alternative explanation for the reduced enzyme activities and associated phenotypes reported for this mutant variant[35,41,42]. Our in vitro (Fig. 3) and in *planta* (Fig. 9) analyses of the substrate binding and catalytic mechanism of the enzyme now provide a reliable set of mutant combinations for engineering phosphatase-dead PPIP5Ks for genetic rescue experiments in various eukaryotes. Our complementation assays in yeast (Supplementary Fig. 9) and in Arabidopsis[10] both suggest that the catalytic activity of the PPIP5K phosphatase domain is dispensable for growth. This is likely due to apparent partial redundancy (biochemical and genetic) of PPIP5K phosphatases with other inositol pyrophosphate phosphatases in yeast[67,68] and in plants[34].

A conserved aspartate (Asp304) has been previously reported as the proton donor in *E. coli* phytase[58] and mutation of Asp304 strongly alters the substrate stereospecificity of this enzyme[60]. In ScVip1$^{PD}$ this aspartate is replaced by glutamate (Glu991), which is conserved among PPIP5Ks from different species, and which adopts different conformations in our apo and substrate-bound structures (Fig. 8a, b). The ScVip1$^{E991A}$ mutant exhibits an overall increase in enzyme activity and reduced stereospecificity (Fig. 8c, d). We hypothesize that both Asp304 in *E. coli* phytase and Glu991 in ScVip1$^{PD}$ influence the size and charge distribution of the substrate binding site, with mutations increasing the processivity of the enzyme at the expense of substrate selectivity.

Mapping previously identified SpAsp1 missense mutations from a genetic suppressor screen onto our ScVip1$^{PD}$ structure revealed that the His686 (His836 in ScVip1$^{PD}$) to Tyr and Cys643 (Cys793 in ScVip1$^{PD}$) to Tyr mutants target the structural Zn$^{2+}$ binding site in PPIP5K (Fig. 1c). This further highlights the contribution of the Zn$^{2+}$ binding site to the structural integrity of the enzyme[69] (Supplementary Fig. 7).

In the case of ScVip1$^{PD}$, a $K_M$ of ~5 μM has been reported for 1-InsP$_7$[25]. Using saturating substrate concentrations, we have derived $K_M$ values of ~70 μM and 45 μM for 1-InsP$_7$ and 1,5-InsP$_8$, respectively, and 1,5-InsP$_8$ is the preferred substrate of ScVip1$^{PD}$ in vitro (Fig. 3a, b). For comparison, the PP-InsP phosphatase Siw14 from yeast has a $K_M$ for 5-InsP$_7$ of ~10 μM[70]. The cytosolic concentration of 1,5-InsP$_8$ in *S. cerevisiae* has been recently estimated at ~0.3 μM[5]. However, the local concentration in proximity to the N-terminal Vip1 kinase domain may be higher.

Our structures reveal that the GAF domain is a distinctive structural feature of PPIP5K phosphatases (Fig. 2a). In phytases, the β1-insert maps to the position of the GAF domain. It has been previously demonstrated that this β1-insert undergoes conformational changes upon InsP$_6$ binding[57] (Fig. 2). Our structures reveal that the PPIP5K GAF domain undergoes much larger conformational changes and plays a pivotal role in substrate binding and in channeling of the substrate to the active site (Figs. 5–7). This is further highlighted by the growth phenotypes of mutant versions of AtVIH2 targeting the GAF domain (Fig. 9). Despite the low degree of sequence conservation, substrate binding, catalysis and regulation appear to be well conserved among PPIP5Ks from different organisms (Fig. 9).

In our 1,5-InsP$_8$ complex structure, one substrate molecule is bound to the outer GAF domain, the second molecule maps close to the active site (Fig. 6a–c). These structures and our mutational analysis define inositol polyphosphates/pyrophosphates as additional ligand candidates for GAF domains[53]. It is noteworthy that a PP-InsP "capture site" outside the active site has also been identified in the PPIP5K kinase domain[45]. Orthophosphate is a known inhibitor of human and yeast PPIP5K phosphatase domains, thereby regulating 1,5-InsP$_8$ levels and contributing to phosphate homeostasis[10,16,40]. We speculate that the inhibitory effect of orthophosphate in ScVip1$^{PD}$ (Fig. 4a) may be caused by the binding of this metabolite to the highly basic substrate channel, where it may block access of the 1,5-InsP$_8$ substrate to the outer substrate binding surface (Fig. 7a, c). We attempted to further characterize this intriguing regulatory mechanism, but ScVip1$^{PD}$ crystals grown in the presence of Pi, sulfate or tungstate did not diffract.

The low-resolution envelope of full-length SpAsp1 and our hydrogen/deuterium exchange mass spectrometry experiments collectively indicate that the PPIP5K kinase and phosphatase domains share only a small interface. The active sites of the kinase and phosphatase domains are situated at a considerable distance from one another, which is incompatible with the hypothesis of direct substrate channeling (Fig. 10c, e). A similar L-shaped arrangement of kinase and phosphatase domains has been previously observed in the crystal structure of fructose-2,6-bisphosphate kinase phosphatase[71]. Consistent with this very open architecture of the enzyme, the phosphatase activity of full-length ScVip1 is not drastically different from that of isolated ScVip1$^{PD}$ (Fig. 10d). However, this appears not to be the case for the kinase activity in human PPIP5K2, where the isolated kinase domain appeared much more active compared to the full-length enzyme[37]. The domain boundaries of the PPIP5K kinase and phosphatase domains suggest the presence of large unstructured N- and C-terminal loops, as well as the occurrence of a large disordered linker region within the α-helical insertion domain the phosphatase module (Fig. 1a, c; Supplementary Fig. 2). These sequence-diverse linker regions may represent protein interaction sites[37,72] that could alter the architecture and regulation of the full-length enzyme. Moreover, the linkers are found phosphorylated in yeast (Supplementary Fig. 2), suggesting that additional regulatory mechanisms of PPIP5K remain to be elucidated.

## Methods

### Expression construct cloning

The kinase (residues 186–522, ScVip1$^{KD}$), phosphatase (residues 536–1107, ScVip1$^{PD}$), loop-truncated phosphatase (residues 536–1107, Δ848-918, ScVip1PD$^{Δ848-918}$), and kinase-phosphatase tandem domains (residues 186–1107, ScVip1$^{KD-PD}$) from *Saccharomyces cerevisiae* Vip1

were cloned from synthetic genes (Twist Bioscience) codon-optimized for expression in *Trichoplusia ni* (BTI-Tnao38) cells in vector pBB3, providing tobacco etch virus protease (TEV) cleavable N-terminal 10xHis and twin StrepII affinity tags (Source Data). Point mutations in the ScVip1 phosphatase domain were generated by site-directed mutagenesis and subcloned in pBB3 (Source Data). For crystallization optimization, the ScVip1$^{PD \Delta848-918}$ construct was engineered by replacing the flexible loop with a short Gly-Ser-Ser-Gly linker. The ScVip1$^{PD}$ $^{\Delta848-918}$ construct was further optimized for co-crystallization with 1,5-InsP$_8$ by replacing Arg547, His548, and Arg551 in the catalytic center with alanines. Full-length Asp1 from *Schizosaccharomyces pombe* was cloned from a synthetic gene codon-optimized for expression in *Escherichia coli* into the pMH-HSSumo vector, providing N-terminal small ubiquitin-like modifier fusion protein (Sumo) containing 7xHis and StrepII affinity tags, as described[40,73].

## Protein expression and purification
ScVip1$^{KD}$, ScVip1$^{PD}$, ScVip1$^{PD \Delta848-918}$, ScVip1$^{PD \Delta848-918 RHR-AAA}$, ScVip1$^{KD-PD}$, and their point-mutated versions were expressed in *Trichoplusia ni* BTI-Tnao38 cells (Boyce Thompson Institute, #160776, Ximbio) using recombinant baculoviruses[74]. Typically, 10 mL of recombinant P3 baculovirus was used to infect 200 mL of BTI-Tnao38 insect cells growing in SF900 media (Gibco) at a cell density of ~2.0 × 10$^6$ cells/mL. The cells were incubated for 72 h at 28 °C with shaking at 110 rpm. Cells were harvested by centrifugation at 1000 × *g* for 20 min at 4 °C when cell viability was 80–85% and GFP fluorescence indicated >50% infection rate. The cell pellets were flash-frozen in liquid N$_2$ and stored at −80 °C in lysis buffer containing 50 mM Bis-Tris HCl (pH 7.5), 500 mM NaCl, protease inhibitor cocktail tablets (cOmplete EDTA-free; Roche), and DNase I (Roche). For protein purification, cells were thawed and lysed by sonication (Branson Sonifier DS450). The lysate was clarified by ultracentrifugation at 55,000 × *g* for 60 min at 4 °C and filtered through 0.45 μm (Durapore Merck) filters. The supernatant was loaded onto a Ni$^{2+}$ affinity column (HisTrap HP 5 mL, Cytiva), washed with 50 mM Bis-Tris HCl (pH 7.5), 500 mM NaCl, 20 mM imidazole, 0.5 mM EDTA, and eluted with wash buffer supplemented with 300 mM imidazole (pH 8.0) directly onto a Strep-Tactin XT 4Flow 5 mL column (IBA). The final elution from the Strep-Tactin column was performed using BXT buffer (IBA) supplemented with NaCl to reach a final concentration of 500 mM. The eluted fractions were incubated with TEV protease for 16 h at 4 °C to remove the 10xHis and twin StrepII affinity tags. The cleaved protein was separated from the tags by a second Ni$^{2+}$ affinity chromatography step. Further purification was achieved by size-exclusion chromatrography on a HiLoad Superdex 200 16/600 pg column (Cytiva) equilibrated in 20 mM HEPES (pH 7.5) and 150 mM NaCl. The peak fractions were concentrated to 5–15 mg/mL and immediately used for crystallization experiments or flash-frozen in liquid nitrogen and stored at −80 °C for enzymatic assays.

The expression and purification protocol for full-length SpAsp1 was derived from the work of Benjamin et al.[40]. Protein expression was done in *E. coli* BL21 (DE3) RIL cells grown in Terrific Broth containing 50 μg/mL kanamycin, 34 μg/mL chloramphenicol, and 1% (v/v) ethanol. When the culture reached an OD$_{600nm}$ of 0.8, protein expression was induced with 0.5 mM isopropyl β-D-1-thiogalactopyranoside (IPTG) at 18 °C for 16 h. Cells were harvested by centrifugation and resuspended in a buffer containing 50 mM Tris-HCl (pH 8.0), 500 mM NaCl, and 10% glycerol, supplemented with a complete protease inhibitor cocktail tablet, DNase I, and 0.5 mg/mL lysozyme (Roth). Cell lysis was achieved by sonication. The lysate was then clarified by centrifugation at 46,500 × *g* for 1 h, and the supernatant was loaded onto a Ni$^{2+}$ affinity column (HisTrap HP 5 mL, Cytiva) equilibrated in 50 mM Tris-HCl (pH 8.0), 500 mM NaCl, and 10% (v/v) glycerol. After a washing step with the same buffer, SpAsp1 was eluted with buffer supplemented with 250 mM imidazole (pH 8.0). The 7xHis-StrepII-SUMO tag was cleaved by treatment with Ulp1 protease during an overnight incubation at

4 °C[75]. The tag-free SpAsp1 protein was separated from the fusion tag by a second Ni$^{2+}$ affinity chromatography step. SpAsp1 was then diluted into 50 mM Tris-HCl (pH 8.0) and 10% (v/v) glycerol to reach a final NaCl concentration of 50 mM and loaded onto a HiTrap Heparin HP 5 mL column (Cytiva) equilibrated with 50 mM Tris-HCl (pH 8.0), 50 mM NaCl, and 10% (v/v) glycerol. SpAsp1 was eluted against a gradient of 50 mM Tris-HCl (pH 8.0), 1 M NaCl, and 10% (v/v) glycerol. Finally, SpAsp1 was applied to a HiLoad Superdex 200 16/600 pg column (Cytiva) equilibrated in 20 mM Hepes (pH 7.5) and 150 mM NaCl. The peak fractions containing SpAsp1 were concentrated and immediately used for negative stain electron microscopy.

## Crystallization and data collection
ScVip1$^{KD}$ (8.0 mg/mL in 20 mM Hepes pH 7.5, 150 mM NaCl) was incubated with 5 mM ADP and 20 mM MgCl$_2$ for 1 h on ice prior to crystallization. Crystals developed in hanging drops composed of 1.0 μL of protein solution and 1.0 μL of crystallization buffer (23 % [v/v] PEG 3,350, 0.1 M citric acid/BIS-Tris propane pH 6.4) suspended over 1.0 mL of the latter as reservoir solution. A complete dataset to 1.18 Å resolution was collected at beam line X06DA of the Swiss Light Source, Villigen, Switzerland (Supplementary Table 1). Triclinic crystals of apo ScVip1$^{PD}$ developed in hanging drops composed of 1.5 μL of protein solution (15 mg/mL in 20 mM Hepes pH 7.4, 150 mM NaCl) and 1.5 μL of crystallization buffer (20 % [v/v] PEG 3,350, 0.2 M NH$_4$NO$_3$). Crystals were derivatized for 18 h in crystallization buffer supplemented with 0.1 mM K$_2$PtCl$_4$ and subsequently snap-frozen in liquid N$_2$ after serial transfer in crystallization buffer supplemented with 20 % (v/v) glycerol as cryoprotectant. A single anomalous diffraction (SAD) dataset at the 'white line' Pt L3 edge was collected at beam line X06DA (Supplementary Table 1). Native crystals of ScVip1$^{PD}$ developed in 20% v/v (+/-)-2-Methyl-2,4-pentanediol, 0.1 M Tris pH 8.0. Crystals were cryoprotected by addition of glycerol to a final concentration of 20% (v/v) and a complete dataset with apparent orthorhombic symmetry was collected at beam line X06DA to 3.05 Å resolution (Supplementary Table 1). Crystals of the engineered ScVip1$^{PD \Delta848-918}$ construct (10 mg/mL in 20 mM Hepes pH7.5, 150 mM NaCl, 0.5 mM (CH$_3$CO$_2$)$_2$Zn) developed in 2 μL drops containing 10 % (w/v) polyvinylpyrrolidone K15, 5 mM CoCl$_2$, 0.1 M Tris pH 7.0 as crystallization buffer and 0.1 μL additive solution (5 mM 13:0 Lyso PC aka 1-tridecanoyl-2-hydroxy-sn-glycero-3-phosphocholine) using microseeding protocols. Crystals were cryoprotected by serial transfer into crystallization buffer containing glyerol to a final concentration of 30% (v/v) and snap frozen in liquid N$_2$. Data collection at beam line X06DA yielded a complete dataset at 3.4 Å resolution (Supplementary Table 1). The ScVip1$^{PD \Delta848-918 RHR-AAA}$–1,5-InsP$_8$ complex crystallized in sitting drops containing 0.25 μL protein solution (12 mg/mL in 20 mM Hepes 7.5, 150 mM NaCl, 2.5 mM 1,5-InsP$_8$) and 0.25 μL of crystallization buffer (Morpheus II screen condition H2, 32.5% v/v Precipitant Mix 6: 25% w/v PEG 4000, 40% v/v 1,2,6-Hexanetriol; 0.04 M Polyamines: 0.01 M Spermine tetrahydrochloride, 0.01 M Spermidine trihydrochloride, 0.01 M 1,4-Diaminobutane dihydrochloride, 0.01 M DL-Ornithine monohydrochloride; 0.1 M Buffer System 4: pH6.5 MOPSO, Bis-Tris)[76]. Crystals were cryoprotected by serial transfer in crystallization buffer supplemented with ethylene glycol to a final concentration of 25 % (v/v) and diffracted up to 2.36 Å resolution (Supplementary Table 1). Data processing was done with XDS[77] (version June 30, 2023).

## Crystallographic structure solution and refinement
The structure of ScVip1$^{KD}$ was solved by molecular replacement method using the program PHASER[78] and the isolated kinase domain of HsPPIP5K2 as search model (https://doi.org/10.2210/pdb3t9a/pdb). The solution comprises a monomer in the asymmetric unit. The structure was completed in alternating cycles of manual model building in COOT[79] and restrained refinement with anisotropic atomic

displacement parameters as implemented in PHENIX.REFINE[80]. The structure of ScVip1[PD] was solved by MR-SAD using a low redundancy SAD dataset collected from a $K_2PtCl_2$-derivatized crystal and a fragment of the *Francisella tularensis* histidine acid phosphatase core domain (https://doi.org/10.2210/pdb3it3/pdb) in the program PHASER-EP[81]. The resulting solution comprised four molecules in the asymmetric unit, four $Pt^{2+}$ and four $Zn^{2+}$ ions (Supplementary Table 1). The starting phases and model were used for non-crystallographic symmetry (NCS) density modification in PHENIX.RESOLVE[82] and the resulting 3.5 Å electron density map was sufficient to build a nearly complete poly-alanine model that was used to determine the structure of ScVip1[PD] apo by molecular replacement. ScVip1[PD] apo crystals had an apparent *C* 2 2 2 symmetry, however no packing solution could be identified with PHASER[78] and analysis with PHENIX.XTRIAGE suggested an abnormal distribution of intensities. The structure was subsequently solved in space group *P* 1 and input into the program ZANUDA[83] for space group validation. ZANUDA returned a solution in space group *P* $2_1$, with four molecules in the asymmetric unit (Supplementary Table 1). The a and c axes are almost identical, with a twin fraction of ~0.5, similar to previous reports[84,85]. The structure was completed in alternating cycles of manual model building and restrained NCS refinement with twin operator l, -k, h in PHENIX.REFINE[80]. A large, apparently disordered loop (residues 851–918) was identified in this structure and subsequently replaced by a short Gly-Ser-Ser-Gly linker. The structures of ScVip1[PD Δ848-918] and ScVip1[PD Δ848-918 RHR-AAA] were solved by molecular replacement using the ScVip1 apo structure as search model and refined with PHENIX.REFINE[80] (Supplementary Table 1). The stereochemistry of the refined models was assessed with PHENIX.MOLPROBITY[86]. Structural diagrams were generated with CHIMERA[87].

## NMR-based phosphatase assay

ScVip1 phosphatase assays were performed as previously described[10]. Reactions contained 100 μM of the respective $[^{13}C_6]$-labeled PP-InsP in 20 mM Hepes pH 7.0, 150 mM NaCl, 0.2 mg/mL BSA, and $D_2O$ to a total volume of 600 μL. Reactions were supplemented with sodium phosphate, sodium sulfate, or sodium nitrate, as indicated. Reaction mixtures were pre-incubated at 37 °C and the reaction was started by adding the respective amount of enzyme (ScVip1[PD] (WT): 10 nM for 1-InsP_7 and 1,5-InsP_8, 700 nM for 5-InsP_7; 1.5 μM R547A-H548A-R551A; 1.25 μM R547A; 1.5 μM H548A; 1.5 μM R551A; 10 nM Δ848-918; 1.5 μM K554A; E991A; 2 μM for InsP_6, 20 nM for 1-InsP_7, 10 nM for 1,5-InsP_8; 1 μM P553V-G646V-G647A; 10 nM C793A; 10 nM H651A; 350 nM K558A-K605A-K732A-K817A; 1 μM K554A-K556A-K644A; 10 nM K817S-D819A-S821A; 50 nM E576A-K623A-Q625S-K627A). Enzyme concentrations were varied in order to monitor the biochemical reactions within the detection range and the time frame of our NMR method. Substrate concentrations were kept at 100 μM, which was required to ensure robust detection by NMR. Turnover of PP-InsPs was monitored continuously at 310 K using a nuclear magnetic resonance spectroscopy (NMR) pseudo-2D spin-echo difference experiment[88]. Individual NMR spectra were recorded on a Bruker AV-III spectrometer (Bruker Biospin) equipped equipped with a cryo-QCI probe operating at 600 MHz for ¹H and 151 MHz for ¹³C nuclei. Measurements and NMR data analysis were performed using TopSpin 3.5. Individual NMR spectra were recorded every 86 s, with a number of scans per spectrum of 128. Spectral width amounted to 16.663 ppm (13.03 ppm to −3.62 ppm). The relative intensity changes of the C2 peaks of the respective PP-InsPs as a function of reaction time were used for quantification[10,88]. Data analysis was carried out with GraphPad PRISM 5. Depending on the degree of reaction progress, kinetic parameters were derived from the respective progress curves using two types of equations. When the reactions were at low conversion (<25%), the reactions were considered to be in the initial range, where the progress curves had a linear

shape. In these cases, the reaction rates were derived by linear regression in GraphPad PRISM. The slope of the graph representing the reaction product (typically 5-InsP_7) corresponded to the non-normalized reaction velocity (unit: μM min⁻¹). This reaction velocity was normalized by dividing the first derivative of the equation by the mass concentration of the enzyme, yielding $v_0$ with the unit nmol mg⁻¹ min⁻¹. If the enzymatic activity and NMR measurement time were sufficient to monitor higher conversions, kinetic parameters were calculated using non-linear regression in GraphPad PRISM. Here, the "one-phase decay " equation $(Y = (Y0 − Plateau)*exp(-K*X) + Plateau)$ was used to fit the hyperbolically shaped progress curve. The non-normalized reaction velocity was calculated from the first derivative of the equation which corresponds to $-K(Y0-Plateau)$. As for linear procress curves, the parameters corresponding to the hydrolysis product (typically 5-InsP_7) were applied to the equation. Again, the non-normalized reaction velocity was subsequently normalized by dividing by the mass concentration of the enyme, yielding $v_0$ with the unit nmol mg⁻¹ min⁻¹. To validate the kinetic parameters calculated with the "one-phase decay " equation, reaction velocities of hyperbolically shaped progress curves were also subjected to linear regression of the initial linear range, fitted using linear regression, and then normalized by dividing by the respective enzyme mass concentration.

In order to determine the substrate preference of ScVip1[PD], reaction mixtures containing 20 mM HEPES pH 7.0, 150 mM NaCl, 0.2 mg/mL BSA and 100 μM of the respective $[^{13}C_6]$-labeled (PP-)InsP were prepared and pre-incubated at 37 °C. ScVip1[PD] was added subsequently (100 nM for 1-InsP_7 and 1,5-InsP_8, 1 μM for InsP_6 and 5-InsP_7) and the reactions were incubated at 37 °C for 3 h. As a control, $H_2O$ was added instead of the enzyme. Reactions were quenched by heating to 95 °C for 5 min. The samples were centrifuged at $15,000 \times g$ for 5 min and the supernatant was analyzed by a¹H-¹³C-BIRD-HMQC 2D NMR as described[88].

## Malachite green-based phosphatase assay

The phosphatase activity of ScVip1[PD] was determined using the Malachite Green Phosphate Assay Kit (Sigma-Aldrich). This assay detects the release of inorganic phosphate catalyzed by ScVip1[PD] from substrates such as 1,5-InsP_8, 1-InsP_7, 5-InsP_7, or InsP_6 during the phosphatase reaction[89,90]. For the assay, 100 nM of ScVip1[PD] was incubated at 37 °C for 2,4, and 8 min in 100 μL reaction mixtures containing 20 mM Hepes (pH 7.5), 150 mM NaCl and varying concentrations of substrates (1,5-InsP_8 ranging from 8 μM to 700 μM; 1-InsP_7 ranging from 1 μM to 2000 μM, unless indicated otherwise). The reaction was quenched by mixing 40 μL of the reaction mixture with 10 μL of malachite green reagent. The color formation was measured after 25 min on a plate reader (Tecan Spark) at 620 nm. When precipitation of the reagent occurred at high phosphate concentrations (>100 μM), the reaction mixture was 2–8 times diluted in buffer prior quenching. All experiments were performed twice (n = 2). The quantity of released inorganic phosphatase was estimated using the standard curve of $OD_{620 nm}$ versus phosphate standard concentrations. Blanks $OD_{620 nm}$ were measured without enzyme at each time point to discard effect of non-enzymatic hydrolysis of PP-InsPs or the eventual presence of inorganic phosphate in the buffer. The kinetic parameters ($v_0$, Vmax, and $K_M$) were calculated using the linear regression analysis and the Michaelis-Menten model implemented in GraphPad PRISM (version 10.3.0.507).

## Thermal shift assay

Thermal shift assays were conducted with 25 to 100 μM of wild-type or mutant ScVip1[PD] in 20 mM Hepes (pH 7.5), 150 mM NaCl, and 8fold dilution of SYPRO Orange dye (Thermo Fisher Scientific). Protein samples were heated with an increasing gradient of 0.05 °C/s from 25 to 99 °C and melting curves were recorded using a QuantStudio 5 Real-Time PCR System system (Thermo Fisher Scientific).

## Analytical size-exclusion chromatography

Gel filtration experiments were performed using a Superdex 200 Increase 10/300 GL column (GE Healthcare) pre-equilibrated in 20 mM Hepes pH 7.5, 150 mM NaCl. 200 µl of the respective protein was loaded sequentially onto the column, and elution at 0.75 ml/min was monitored by ultraviolet absorbance at 280 nm. Peak fractions were analyzed by SDS–PAGE gel electrophoresis.

## Yeast strains and plasmids

The *Saccharomyces cerevisiae* strains used in this study are listed in Source Data. Genomic disruptions at genomic loci were performed as previously reported[91]. Preparation of media, yeast transformations, and genetic manipulations were performed according to established protocols. The plasmids and primers used for site-directed mutagenesis used in this study are listed in Source Data. All recombinant DNA techniques were performed according to established procedures using Escherichia coli TOP10 cells for cloning and plasmid propagation. Gene mutations were generated with QuickChange site-directed mutagenesis kit (Agilent Technologies, Santa Clara, CA, USA). All cloned DNA fragments and mutagenized plasmids were verified by Sanger-sequencing (Microsynth, AG).

## Molecular cloning and generation of stable transgenic A. thaliana lines

The Golden Gate system was used to generate plasmids for Arabidopsis transformation[92]. Mutant versions of the AtVIH2 coding sequence were synthesized (Twist bio-science, San Francisco, CA) (Source Data). The different mutants were then cloned into the level one (L1) vector (Source Data). To construct the binary vector L1 vectors containing the AtUBI10 promoter (proUBI10), the corresponding AtVIH2 coding sequence, a FLAG tag, the Nos terminator, and Fast Red as a fluorescent marker for the seed coat. The plasmids were transformed into *A. tumefaciens* strain GV3101. The floral dip method was then used to transform 5-week-old plants (Col-0)[93]. T1 plants were selected by red fluorescence using a Nikon SMZ18 stereomicroscope equipped with an RFP-B filter. Lines with single T-DNA insertions were selected by the segregation analysis. All experiments were performed using stable T3 generation lines (Source Data).

## Plant material, seed sterilization, and plant growth conditions

Seeds were sterilized using the chloride gas protocol. Seeds were then sown on ½MS media containing 1% (w/v) sucrose, MES (0.5 g/L), pH 5.7 and agar (8 g/L). 120 mm square Petri dishes were incubated for 48 h at 4 °C in the dark and transferred to a growth chamber at 22 °C with a 16 h/8 h light/dark cycle under a fluorescent light source. 7 d old seedlings were transferred to soil in 8 cm pots and grown as indicated.

## Image acquisition and rosette area quantification

To evaluate the rosette phenotypes of the different T3 lines, 16 plants (in 4 pots) were analyzed. After 2 weeks in soil, top images were recorded and the rosette area was manually segmented in Fiji[94]. Specifically, images were converted to 8 bit followed by manual selection of individual plants. Contrast was adjusted, and the rosette was selected using the threshold. The scale (240 px to 1.5 cm) was set before quantification. Finally, the area of the segmented image was determined using the Particle Quantify function as implemented in Fiji.

## Determination of cellular Pi concentration

For Pi quantification 3-weeks-old plants were analyzed (8 independent plants per sample). One cotyledonous leaf per plant was harvested, the fresh weight was recorded and the leaf was transferred to an Eppendorf tube (1.5 ml) containing 500 µl miliQ $H_2O$. The tubes were then frozen in liquid nitrogen and boiled at 80 °C for 5 min. This step was repeated two times. Measurement was performed using a colorimetric molybdate assay[89]. The master solution contains contained 72 µL of ammonium molybdate solution (0.0044 % [w/v] of ammonium molybdate tetrahydrate, 0.23 % [v/v] of 18 M $H_2SO_4$), 16 µL of 10 % (w/v) acetic acid and 12 µL of miliQ $H_2O$. For each reaction, 20 µl of sample were added in triplicate to a 96-well plate and incubated with 100 µl of working solution for 1 h at 37 °C. Absorbance was measured at 820 nm in a Spark plate reader (Tecan). The concentration was determined according to the standard curve with a concentration range of 1, 0.5, 0.2, 0.1, and 0 mM $Na_2HPO_4$.

## Western blotting

The recombinant expression of 10xHis-tagged AtVIH1[PD], AtVIH2[PD] and MpVIP1[PD] phosphatase domains in baculovirus-infected BTI-Tnao38 insect cells was assayed by western blotting. Typically, 40 mL of recombinant P2 baculovirus was used to infect 200 mL of BTI-Tnao38 insect cells growing in SF900 media (Gibco) at a cell density of ~1.5 × 10⁶ cells/mL. The cells were incubated for 72 h at 28 °C while shaking at 110 rpm. Cells were harvested by centrifugation at 1000 × $g$ for 20 min at 4 °C when cell viability was 80–85% and GFP fluorescence indicated >50% infection rate. Proteins were extracted from ~4 g of cell pellet, resuspended in 20 mM Hepes pH 7.5, 500 mM NaCl, 3 mM β-mercaptoethanol, supplemented with a cOmplete protease inhibitor cocktail tablet and DNase I. Cells were lysed by sonication and the lysate was clarified by ultracentrifugation at 55,000 × $g$ for 60 min at 4 °C. The pellet, solubilized in 8 M urea, and the supernatant, were transferred in 30 µl of SDS loading buffer and incubated at 95 °C for 5 min. A 10% BIS-Tris acrylamide SDS-PAGE was run with the MES running buffer. Proteins were transferred to a 0.2 µm PVDF membrane (IB34002, Thermo Scientific™) using a semi-dry protocol (iBlot3, Thermo Scientific). Membranes were blotted with TBS-Tween (0.1%) - milk (5%) for 1 h at room temperature. For 10xHis tag detection, membranes were incubated with a Anti-His6-Peroxidase (11 965 085 001 Merck/Sigma-Aldrich), mouse monoclonal antibody (clone BMG-His-1, LOT 16830100) using a dilution of 1:3000. Detection was performed using SuperSignal West Femto Maximum Sensitivity Substrate (34095, Thermo Scientific).

Proteins were extracted from 3-weeks-old plants. About 100 mg of sample were collected in 2 ml Eppendorf tubes and frozen in liquid nitrogen. The tissue was homogenized in a tissue lyzer (MM400, Retsch). Then, 300 µl of 1x PBS buffer supplemented with the plant-specific protease inhibitor cocktail (PE0230, Merck) was added to each tube and incubated for 30 min at 4 °C in an orbital shaker (Intelli-Mixer RM-2M, ELMI). The samples were then centrifuged at 10,000 × $g$ for 20 min at 4 °C. Then, 150 µl of the supernatant was transferred in a new tube containing 30 µl of SDS loading buffer and incubated at 95 °C for 5 min. A 10% BIS-Tris acrylamide SDS-PAGE was run with the MES running buffer. Proteins were transferred to a 0.2 µm PVDF membrane (IB34002, Thermo Scientific) using a semi-dry protocol (iBlot3, Thermo Scientific). Membranes were blotted with TBS-Tween (0.1%) - milk (5%) for 1 h at room temperature. For FLAG detection, membranes were incubated with a monoclonal Anti-FLAG-M2-Peroxidase (HRP) antibody (A8592-5X1MG Merck/Sigma-Aldrich, clone M2, LOT # SLBH1183V) using a dilution of 1:5000. Detection was performed using SuperSignal West Femto Maximum Sensitivity Substrate (34095, Thermo Scientific).

## Electron microscopy and image processing

The full-length SpAsp1 envelope was analyzed using negative-stain electron microscopy. A total of 5 µl of monomeric peak fractions from a gel filtration run (concentration 0.01 mg/mL) were applied to glow-discharged, carbon-coated copper grids (400 mesh, Electron Microscopy Sciences). After incubation for 1 min, excess liquid was blotted off. The grids were then sequentially passed through one drop of buffer (20 mM Hepes, pH 7.5, and 150 mM NaCl) and two drops of 2% (w/v) uranyl acetate solution, with the sample incubated in the last drop for 1 min before blotting and air drying. A total of 1,565

micrographs were recorded at the DCI-Geneva (cryoGEnic) electron microscopy platform using a Talos L120C microscope operating at 120 kV and equipped with a Falcon II direct electron detector. Data acquisition was automated using EPU software (Thermo Fisher Scientific) with a pixel size of 1.531 Å and a total electron dose of 52 e-/Å². Data processing was performed using CryoSPARC (version 4.1)[95]. Initially, contrast transfer function (CTF) parameters were estimated using patch-based CTF estimation. Micrographs with a CTF fit better than 10 Å resolution were selected, resulting in 1497 micrographs. Manual picking of 177 SpAsp1 particles, which were extracted in boxes of 250 × 250 pixels, were used to train the blob picker tuner. After this process, 239 exposures were rejected due to containing either too few or excessively high numbers of picked particles, leaving a particle stack of 270,221 particles (averaging 220 particles per micrograph) with defocus values between 0.3 and 1.3 μm. Two rounds of reference-free 2D class averaging were employed to further refine the dataset and remove the stain artifacts, resulting in 107,485 particles. An ab initio reconstruction with six classes was followed by 2D classification and non-uniform refinement for each class. The best 3D reconstruction, representing full-length SpAsp1, comprised 11,837 particles.

## Hydrogen/deuterium exchange mass spectrometry

HDX-MS experiments were performed at the UniGe Protein Platform (University of Geneva, Switzerland) following an established protocol with minimal modifications[96]. Details of reaction conditions and all data are presented in Supplementary Fig. 11. HDX reactions were performed in 50 μl volumes with a final protein concentration of 2.1 μM of ScVip1 and a 100-fold molar excess of AMP-PNP and/or PCP-IP$_8$[97]. Briefly, 107 picomoles of protein were pre-incubated with ligands for 1 h on ice in a final volume of 7 μl. The deuterium on-exchange reaction was initiated by adding 43 μl of D$_2$O exchange buffer (10 mM Tris pH 8/150 mM NaCl in D$_2$O) to the protein-ligand mixture. Reactions were carried-out at room temperature for 2 incubation times (30 s, 300 s) and terminated by the sequential addition of 20 μl of ice-cold quench buffer (4 M Gdn-HCl/1 M NaCl/0.1 M NaH$_2$PO$_4$ pH 2.5/1 % (v/v) formic acid). Samples were immediately frozen in liquid nitrogen and stored at −80 °C for up to 2 weeks. All experiments were repeated in triplicate ($n = 3$). To quantify deuterium uptake into the protein, samples were thawed and injected in a UPLC system immersed in ice with 0.1 % FA as liquid phase. The protein was digested via two immobilized pepsin columns (Thermo Scientific #23131), and peptides were collected onto a VanGuard precolumn trap (Waters). The trap was then eluted, and peptides were separated with a C18, 300 Å, 1.7 μm particle size Fortis Bio 100 × 2.1 mm column over a gradient of 8–30 % buffer C over 20 min at 150 μl/min (buffer B: 0.1 % [v/v] formic acid; buffer C: 100 % acetonitrile). Mass spectra were acquired on an Orbitrap Velos Pro (Thermo), for ions from 400 to 2200 m/z using an electrospray ionization source operated at 300 °C, 5 kV of ion spray voltage. Peptides were identified by data-dependent acquisition of a non-deuterated sample after MS/MS and data were analyzed by Mascot 2.6 using a database composed of purified proteins and known contaminants. Precursor mass tolerance was set to 10 ppm and fragment mass tolerance to 0.6 Da. Protein digestion was set as non-specific. All peptides analyzed are shown in Supplementary Fig. 11. Deuterium incorporation levels were quantified using HD examiner software version 3.3 (Sierra Analytics), and quality of every peptide was checked manually. Results are presented as percentage of maximal deuteration compared to theoretical maximal deuteration level. Changes in deuteration level between two states were considered significant if >7% and >0.5 Da and $p < 0.02$ (unpaired $t$-test) for a single deuteration time.

## Reporting summary

Further information on research design is available in the Nature Portfolio Reporting Summary linked to this article.

## Data availability

The data that support this study are available from the corresponding authors upon request. Crystallographic coordinates and associated structure factors have been deposited with the Protein Data Bank (PDB): 9GR8 (ScVip1$^{KD}$−AD); 9GRH (ScVip1$^{PD}$−apo); 9GRN (ScVip1$^{PD\ \Delta 848-918}$−apo) and 9GRO (ScVip1$^{PD\ \Delta 848-918\ RHR-AAA}$−1,5-InsP$_8$). The mass spectrometry proteomics data have been deposited to the ProteomeXchange Consortium via the PRIDE partner repository[98] with dataset identifier PXD056020. The source data underlying Figs. 3a, b, d, e, 4a–c, 5c–e, 7e, f, 8c, d, 9c, d, 10d, e, and Supplementary Figs. S7c, S10b, c are provided as a Source Data file. Source data are provided with this paper.

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

## Acknowledgements

We thank P. Rieu and A. Caregnato for critical reading of the manuscript. This work was supported by Sinergia grant CRSII5_209412 from the Swiss National Science Foundation (to D.F., V.G.P., and M.H.) and by a Howard Hughes Medical Institute International Research Scholar Award 55008733 (to M.H.).

## Author contributions

P.R. and K.L. expressed and purified proteins, K.L. and P.R. crystallized proteins and solved and refined structures with the help of M.H. P.R. and S.M.B. performed and analyzed enzyme assays with the help of D.F. D.F. contributed PP-InsP reagents. P.R. performed thermal shift assays and size-exclusion chromatography experiments. D.P-C. and V.G.P. designed and performed yeast experiments. F.R. performed plant genetic experiments, Pi measurements and western blotting. P.R. performed and analyzed negative stain electron microscopy experiments. K.L. and O.V. performed and analyzed HDX-MS experiments. P.R. and M.H. wrote the manuscript. All authors edited the manuscript and approved the final document.

## Competing interests

The authors declare no competing interests.
