## [Transparent Peer Review file · Nature Communications]

A small signaling domain controls PPIP5K phosphatase activity in phosphate homeostasis

Corresponding Author: Professor Michael Hothorn

Version 0:

Reviewer comments:

Reviewer #1

(Remarks to the Author)

This study provides a comprehensive characterization of the enzyme PPIP5K, a pivotal component in the metabolism of inositol polyphosphates, specifically PP-InsPs.

The work presents several structural versions of ScPPIP5K, including both kinase and phosphatase domain structures. Uncovering the phosphatase domain's structure for the first time is particularly significant, as it fills a crucial knowledge gap in the field. To achieve this, the authors managed to crystallize an engineered phosphatase domain expressed in insect cells. This domain presents a canonical histidine-acid phosphatase with two insertions: a GAF-like domain, distinctive of PPIP5K, and an alpha-helical domain, both important for enzymatic function. Notably, the authors identify two substrate binding sites and a channel formed between them, claimed to be the substrate entrance to the active site. Here, the GAF domain moves from an open to a closed conformation, and the authors hypothesize on the determinants of GAF movement. They also validate residues involved in catalytic activity and substrate stereospecificity, detect a direct enzymatic relation to Pi (an inhibitor of the enzyme), find a Zn binding site involved in enzymatic stability, and convincingly demonstrate the independence of the kinase and phosphatase domains through structural and functional assays.

This manuscript offers robust data through various experimental approaches, including X-ray crystallography, NMR, HDX-MS, negative-stain electron microscopy, in combination with a meticulous mutagenetic analysis, in vitro activity/kinetics measurements, and results with plant and yeast, which provide an in vivo perspective on the enzyme's functionality. The combination of structural and functional techniques provides a thorough exploration of PPIP5K activity and structural configuration. Each method is explained with sufficient detail, allowing for reproducibility.

The manuscript is well-organized and clearly presented. The high-quality figures and diagrams effectively communicate the structural and functional findings, making complex information accessible. The text is appropriately contextualized with previous literature, supporting the relevance of this work within the field.

This substantial discovery significantly enhances our understanding of this enzyme's function. This research marks a notable breakthrough in the structural biology of inositol polyphosphate metabolism. In my opinion, this manuscript is an excellent piece of work and should be accepted as it is.

Here are a few minor comments for authors:

- A brief explanation of the catalytic mechanism in His-acid phosphatases, in the introduction, could be helpful.
- I am not sure that the term "domain swapped" (line 212, page 7) is properly used in the manuscript. Based on Fig. S8, two PPIP5K subunits form a disulfide bond, but there does not seem to be a domain swap involving an exchange of secondary structure elements between domains.
- In Figure 4c, the legend would be easier to understand with additional explanation. For example: InsP = 1,5-InsP8 or InsP=5-InsP7.

Reviewer #2

(Remarks to the Author)

Summary

In this work, the authors present structural and functional insights regarding the phosphatase domains inherent to PPIP5K inositol pyrophosphate kinase/phosphatases. These domains have long remained an elusive subject in the field of inositol-pyrophosphate signaling, as they contain long intrinsically disordered regions which resist soluble recombinant expression. Remedying this, the authors have engineered versions of yeast PPIP5K phosphatase domains with these IDRs replaced by a short linker sequence. Data reported in this work from the crystallization of these proteins highlight important enzyme features conserved across many eukaryotes, including a structural Zinc binding site and a GAF domain-induced substrate channel. Enzymatic kinetic studies presented in this work characterize the substrate specificity of these enzymes, with a

strong affinity for the 1,5-InsP8 isoform. Further, the authors demonstrate a mechanism by which free cellular orthophosphate inhibits substrate binding. Together, these data constitute a dramatic step forward in the field's understanding of eukaryotic PPIP5K phosphatase structure and function.

Remarks

I have no objections to this manuscript in its current form. The data are well presented, concise, and methodologies are well described. The authors deposit these data to publicly available repositories, on hold for publication. The methods presented are sufficient for replication, and provide clear technical detail.

2. Response to minor comments from Reviewer #1

- A brief explanation of the catalytic mechanism in His-acid phosphatases, in the introduction, could be helpful.

###

Our response:

Thank you for pointing this out to us. We have added the following statement in the introduction section of our revised manuscript (lines 101-103): "Histidine acid phosphatases catalyze the hydrolysis of phosphomonoesters via a two-step mechanism. First, the conserved histidine acts as a nucleophile, forming a covalent phospho-histidine intermediate. In the second step, a water molecule hydrolyzes the intermediate, releasing free inorganic phosphate⁴⁸."

In light of our finding that ScVip1PD is a specific inositol pyrophosphate 1-phosphatase acting on a phosphoanhydride, not a phosphoester bond, we have added this statement to the revised discussion section (lines 346-347): " It is noteworthy that ScVip1^{PD} specifically catalyzes the hydrolysis of a phosphoanhydride bond, not of a phosphomonoester as seen with other histidine acid phosphatases (Fig. 3). "

###

- I am not sure that the term "domain swapped" (line 212, page 7) is properly used in the manuscript. Based on Fig. S8, two PPIP5K subunits form a disulfide bond, but there does not seem to be a domain swap involving an exchange of secondary structure elements between domains.

###

Our response:

Indeed, the crystallographic dimer does not represent a case of domain swapping. The revised statement in the result section reads (lines 215-217): "... we located a crystallographic dimer stabilized by an intermolecular disulfide bridge involving a cysteine (Cys793) that, in apo structures, is part of the zinc-binding site (Supplementary Fig. 8)."

###

- In Figure 4c, the legend would be easier to understand with additional explanation. For example: InsP = 1,5-InsP8 or InsP=5-InsP7.

###

Our response:

Changed as requested.

###